# Changes in diazotrophic community structure associated with Kuroshio succession in the northern South China Sea

Han Zhang[1,2], Guangming Mai[1], Weicheng Luo[2], Meng Chen[1], Ran Duan[2,§], Tuo Shi[1,2]

[1] Marine Genomics and Biotechnology Program, Institute of Marine Science and Technology, Shandong University, Qingdao, Shandong, PR China
[2] State Key Laboratory of Marine Environmental Science, College of Ocean and Earth Sciences, Xiamen University, Xiamen, Fujian, PR China
[§] Present address: Department of Marine Science, University of Southern California, Los Angeles, USA
*Correspondence to*: Tuo Shi (tuoshi@sdu.edu.cn)

**Abstract** Kuroshio intrusion (KI) is a key process that transports water from the Western Pacific Ocean to the northern South China Sea (nSCS), where KI-induced surface water mixing often causes variations in microbial assemblages. Yet, how interannual KIs affect biogeography of diazotrophs and associated environmental factors, remains poorly characterized. Here, by quantifying the degree of KIs in two consecutive years, coupled with monitoring the diversity and distribution of nitrogenase-encoding *nifH* phylotypes with quantitative PCR and high-throughput sequencing, we show that changes in the diazotrophic community structure in the nSCS are highly correlated with KI leading to variations in a range of physicochemical parameters. Specifically, the filamentous cyanobacterium *Trichodesmium* was more abundant at stations strongly affected by KI, and hereby with deeper mixed layer, higher surface salinity, and temperature; the unicellular $N_2$-fixing cyanobacteria in group B (UCYN-B) were more abundant at stations least affected by KI and correlated with nutrient availability, whereas UCYN-C and the γ-proteobacteria were prevalent at stations moderately affected by KI. Neutral community model further demonstrated that dominant diazotrophic subcommunities were significantly affected by environmental factors in 2017 when KI was stronger compared to 2018 when KI retreated. Our analyses provide insightful evidence in the role of KI succession in shaping diazotrophic community structure primarily as a stochastic process, implying a potential region-scale redistribution of diazotrophs and nitrogen budget, given that KIs are projected to intensify in a future warming ocean.

**Keywords**: Biogeography, Kuroshio, Marine diazotroph, *nifH* phylotype, South China Sea

# 1 Introduction

Biological nitrogen fixation, the process in which a specialized group of prokaryotes called diazotrophs convert atmospheric nitrogen ($N_2$) to ammonia ($NH_3$), provides one of the primary nutrients to the biosphere (Gruber, 2008). In the oligotrophic ocean, marine $N_2$ fixation is an important source of new nitrogen (N) to the euphotic zone, providing up to 50% of bioavailable N to marine phytoplankton (Wang et al., 2019) in sustaining net carbon sequestration and export of organic carbon to the deep ocean, and potentially governing primary productivity in geologic time scales (Falkowski et al., 1998; Karl et al., 2002). Marine $N_2$ fixation is, therefore, of significant importance to global carbon cycle (Falkowski et al., 2000; Zehr and Capone, 2020; Hutchins and Capone, 2022).

The field of marine $N_2$ fixation is changing dramatically. Based on the polymerase chain reaction (PCR) technique amplifying *nifH* gene encoding the catalytic subunit of nitrogenase (Zehr et al., 1998), a number of marine planktonic diazotrophs have been identified. The bloom-forming filamentous cyanobacteria, including the free-living non-heterocystous *Trichodesmium* (Capone, 1997; Capone et al., 2005) and the heterocystous *Richelia* and *Calothrix* symbiotically associated with diatoms (Villareal, 1992; Foster et al., 2007) were traditionally recognized to be the prevalent marine diazotrophs. This previously established concept was challenged by the discovery of unicellular, $N_2$-fixing cyanobacteria (UCYN) belonging to *Crocosphaera*, *Cyanothece* and other closely related genera, which fix equivalent or even higher amount of $N_2$ globally relative to the summed contribution by *Trichodesmium* and the diatom-associated symbionts (Zehr et al., 2001; Montoya et al., 2004; Martínez-Pérez et al., 2016). Phylogenetically distinct clades of UCYN have been detected. The "*Candidatus* Atelocyanobacterium thalassa" in group A (UCYN-A) appears to be an obligate endosymbiont of the single-celled prymnesiophyte algae (Thompson et al., 2012), whereas UCYN-B is mostly free-living and a culturable representative, *Crocosphaera watsonii,* has been successfully isolated. Compared with other major $N_2$ fixers like *Trichodesmium* and *C. watsonii*, the UCYN-A can be detected in colder and deeper open ocean waters (Moisander et al., 2010), as well as coastal waters (Hagino et al., 2013), further demonstrating their importance in global $N_2$ fixation. The advent of global-scale metagenomic databases has facilitated significant advances in the study of $N_2$ fixation (Muñoz-Marín et al., 2019; Acinas et al., 2021; Pierella Karlusich et al., 2021; Dong et al., 2022). Particularly, diazotrophs genomes reconstructed from global metagenomic data have extended the PCR-based amplicon surveys revealing new diatom-diazotroph symbioses (Schvarcz et al., 2022) and new species of non-cyanobacterial diazotrophs (NCDs) (Bombar et al., 2016; Delmont et al., 2018).The Rhopalodiaceae diatoms have been repeatedly isolated from the subtropical North Pacific Ocean, with the endosymbionts having *nifH* gene sequences similar to those of free-living UCYN-C cyanobacteria (Schvarcz et al., 2022). NCDs have been reported to be ubiquitous in ocean ecosystems and contribute to global marine $N_2$ fixation (Moisander et al., 2014; Chakraborty et al., 2021; Turk-Kubo et al., 2022). Recently, widespread and high $N_2$ fixation was also discovered in the coastal or continental shelf (Tang et al., 2019).The South China Sea (SCS) is the largest semi-enclosed marginal sea in the tropical Western Pacific Ocean (WPO). The main water exchange between SCS and WPO occurs as Kuroshio, an energetic western boundary current of the North Pacific intrudes into the SCS through the Luzon Strait (Hu et al., 2015). The intruded Kuroshio splits into the anticyclonic current loop west of the Luzon Strait (i.e., the looping path), the westward South China Sea Branch of the Kuroshio on the slope (i.e., the leaking path), and the cyclonic eddy west of the Luzon Island (i.e., Luzon loop) (Fig. 1) (Nan et al., 2011). Thus, Kuroshio intrusion (KI) plays a significant role in affecting the temperature, salinity, circulation, and the generation of mesoscale eddies and internal waves in the northern SCS (nSCS) (Hu et al., 2000; Yang et al., 2014; Nan et al., 2015). The physical processes of KI have been documented to influence the community structure of bacteria (Zhang et al., 2014), picoplankton (Li et al., 2017), microphytoplankton (Wang et al., 2020b), and microzooplankton (Sun et al., 2021) in the nSCS. Moreover, the transport of diazotrophic lineages such as *Trichodesmium* (Chen et al., 2003) and the lateral input of dissolved organic matter by KI (Xu et al., 2018) also influence the nutrient biogeochemistry of the euphotic zone (Du et al., 2013), which is known to enhance both the

phytoplankton productivity (Chen et al., 2008) and export production (Nishibe et al., 2015; Huang et al., 2019) in the nSCS. In this regard, the nSCS provides an ideal region for studying the dynamics in diazotrophic communities and their activity associated with KI.

The diversity, abundance, and $N_2$ fixation rate of the diazotrophic community in both the SCS and Kuroshio have been well studied (Voss et al., 2006; Moisander et al., 2008; Zhang et al., 2011; Liu et al., 2020; Ding et al., 2021). Much higher diazotroph abundances and $N_2$ fixation rates were found in the Kuroshio current than those in the nSCS (Chen et al., 2014; Cheung et al., 2017; Wu et al., 2018; Lu et al., 2019). Studies also showed that *Trichodesmium* is by far the most abundant cyanobacterial diazotroph (Moisander et al., 2008; Chen et al., 2011; Shiozaki et al., 2014; Xiao et al., 2015) and the indicator species of Kuroshio

(Shiozaki et al., 2015b). However, it was recently suggested that $N_2$ fixation by UCYN is more significant (Wu et al., 2018; Li et al., 2021) whereas the diatom-associated symbionts and NCDs only constitute a small proportion of the diazotrophic community in the Kuroshio (Cheung et al., 2017). In contrast to the well studied diazotrophic biogeography in these regions, less effort has focused on the impacts of KI on marginal seas of the northwestern Pacific (Jiang et al., 2023). Furthermore, the distribution pattern of lineage-specific diazotrophs in the nSCS in relation to KI, remains poorly characterized. For a better understanding of the

physicochemical processes controlling diazotrophic activity and community in response to KI, an ecological survey focusing on the composition and distribution patterns of diazotrophs in the nSCS during Kuroshio succession is necessary. Moreover, considering the annual and interannual variabilities in the intensity and frequency of KI driven primarily by the East Asian monsoon (Cai et al., 2020), there is also an urgent need to assess the impacts of KI as a solo stochastic process (e.g., current-induced drift) or coupled with other extreme climate events (e.g., tropical storms, marine heatwaves).

We hypothesize that KI reshapes diazotrophic communities and the changes are proportional to KI intensities. To test this hypothesis, we carried out ecological surveys on the diversity, abundance, and distribution of diazotrophic communities in two consecutive cruises in the nSCS during summer of 2017 and 2018. By examining the relative abundance of targeted and all-inclusive diazotrophic groups with the *nifH* amplicons, along with measurements of environmental parameters and modeling the contribution of KI into the nSCS, we identified a pattern in the composition and distribution of diazotrophic community that is

highly correlated with Kuroshio succession. We further assessed the role of KI in shaping diazotrophic community structure in the context of stochastic processes using a neutral community model (NCM), demonstrating that the correlation between diazotrophic community assembly and KI is tightly regulated by environmental factors. Taken together, our analyses provide novel mechanistic insight into the biogeography of marine diazotrophs in the nSCS, particularly as it relates to KI.

## 2 Materials and Methods

### 2.1 Sampling and collection of environmental parameters

Sampling for this study was conducted during two research cruises to the SCS aboard the research vessels 'Dong Fang Hong 2' from August 4 to 17, 2017 and 'Tan Kah Kee' from June 12 to July 14, 2018, respectively. Samples were collected using a Seabird SBE 911Plus rosette sampling system (Sea-Bird Electronics, USA) at 14 stations (Fig. 1 and Table S1), seven of which were chosen for both daytime and nighttime sampling, whereas the other seven stations were used only for either daytime or nighttime sampling.

At each station, 1.5–3 L of seawater was prefiltered through a 100-μm pore-size nylon mesh to remove large zooplankton and fish, and then filtered through a 0.22-μm pore-size 47-mm diameter polycarbonate membrane (Millipore, USA) with low pressure (<100 mm Hg pressure) for subsequent DNA extraction. The membranes were flash frozen in liquid nitrogen and stored at –80℃ in the laboratory until further processing. Temperature, salinity, and depth data were recorded using the Seabird system. Light attenuation was determined using a PRR-600 profiling reflectance radiometer (Biospherical Instruments Inc., USA) (Shiozaki et al., 2015a).

Seawater samples for the inorganic nutrients and chlorophyll $a$ (Chl $a$) analysis were prefiltered through GF/F filter membranes (Whatman, USA) and then collected into 100-mL HCl-rinsed polyethylene bottles and refrigerated at –20°C during the cruise and processed immediately after returning to the laboratory. Nitrate ($NO_3^-$) and phosphate ($PO_4^{3-}$) were estimated using Technicon AA3 Auto-Analyzer (Bran Lube, Germany). The detection limits of $NO_3^-$ and $PO_4^{3-}$ concentrations were 0.1 μM and 0.08 μM, respectively. Chl $a$ was extracted with 90% acetone and measured using a Trilogy fluorometer (Turner-Designs, USA).

## 2.2 Estimation of KI fractions

To quantify the hydrographic changes induced by lateral KI, an isopycnal mixing model (Du et al., 2013) was applied to calculate the contribution of KI at each individual station. In the isopycnal model, two water mass endmembers are required to index their relative contribution when they are mixed in different proportions. Here we selected the South-East Asian Time-series Study (SEATS) station (stn9) at 116°E/18°N to represent the proper SCS water mass endmember and the station stn1 at 123.023°E/22.016°N as the Kuroshio water mass endmember. This model assumes that diapycnal mixing is negligible when compared with isopycnal mixing, which allows us to quantify the relative contribution of the SCS ($R_S$) and the Kuroshio ($R_K$) water mass for observed water parcels in the temperature ($\theta$)–salinity ($S$) diagram, based on the conservation of either $\theta$ or $S$ along an isopycnal surface. While Kuroshio can intrude down to 400 m in the SCS, we focused only on the euphotic zone, which typically extends from the surface to 100-m water depth in the nSCS (Chen et al., 2008), where biological alterations of nutrients are highest and may result in profound changes in primary productivity. The averaged Kuroshio fraction in the upper 100 m for a given station ($R_{K\_100}$, %) was therefore employed as a proxy of KI index. The stations with $R_{K\_100}$>75% were considered strongly affected by KI, and those with 25%≤$R_{K\_100}$≤75% and $R_{K\_100}$<25% were considered moderately and least affected by KI, respectively.

## 2.3 Measurements of N$_2$ fixation and primary production

Nitrogen fixation rates (NFR) were assessed using the $^{15}N_2$ stable isotope method (Montoya et al., 1996) with modifications to avoid problems of bubble dissolution (Mohr et al., 2010; Großkopf et al., 2012). Briefly, 5 mL of 98% pure $^{15}N_2$ gas (Cambridge Isotope Laboratories, USA) was injected into an acid-washed, gas-tight plastic bag containing 500 mL of degassed seawater prepared according to Shiozaki et al. (2015a). The bag was gently tapped until the gas was dissolved completely. Sixty mL of $^{15}N_2$-enriched seawater was added into each of the triplicate acid-washed 2.3-L polycarbonate bottles. NaH$^{13}CO_3$ (99 atom% $^{13}C$, Cambridge Isotope Laboratories, USA) solution was added at a final tracer concentration of 100 μM. Each bottle was covered with neutral-density screen to adjust the levels of illumination to match those at the sampling depths. Light conditions were recorded at 5 fixed depths (5, 25, 50, 60–75, 115–150 m) within the water column, corresponding to approximately 50%, 30%, 10%, 1%, and 0.1% of the surface irradiance, respectively. To maintain approximate *in situ* temperatures, the bottles were placed in an on-deck incubator that was connected to a cooling system and continuously flushed with surface seawater. After 24 h of incubation, the samples were filtered (vacuum pressure <100 mm Hg) onto pre-combusted (450°C, 4 h) 47-mm diameter GF/F membranes, and the membranes were immediately stored at −80°C until further processing. To estimate the natural and tracer-enriched $^{15}N$ and $^{13}C$ abundance, the samples were first acid fumed to remove the inorganic carbon and then analyzed using a Delta V Plus isotope ratio mass spectrometry interfaced with a Flash HT 2000 elemental analyzer (Thermo Fisher Scientific, Waltham, MA, USA). The NFR and primary production were determined according to Montoya et al. (1996) and Hama et al. (1983), respectively. The detection limit for N$_2$ fixation rates was estimated by taking 4‰ as the minimum acceptable change in the $\delta^{15}N$ of particulate nitrogen (i.e., a change of 0.00146 in the $^{15}N$ enrichment of particulate nitrogen). The depth-integrated N$_2$ fixation ($I_{NFR}$) and primary production ($I_{PP}$) were calculated by integrating the rates at three to five water depths in the upper 100 m using the trapezoidal integration

method (Wen et al., 2022).

**2.4 DNA extraction**

For one piece of sample membrane, DNA was extracted with the cetyltrimethylammonium bromide (CTAB) method (Yuan et al., 2015) in 600 μL DNA lysis buffer (10 mM Tris-HCl, pH 8.0; 100 mM EDTA, pH 8.0; 0.5% [w/v] SDS) containing 10 mg mL$^{-1}$ Proteinase K (Roche, USA). The extracted DNA was purified using the DNA Clean & Concentrator kit (Zymo, USA) and the concentrations of DNA were measured with the Qubit fluorometer 3.0 (Thermo Scientific, USA) using the broad-range DNA assay kit (Invitrogen, USA). All the DNA samples were eluted in 50 μL DNase/RNase-free ultrapure water (Invitrogen, USA), and stored at –80°C until further processing.

**2.5 TaqMan qPCR assay of targeted *nifH* phylotypes**

The DNA samples were used as templates to quantify the *nifH* gene copies using TaqMan qPCR technique targeting 10 major diazotrophic phylotypes, including the filamentous *Trichodesmium*, the UCYN in groups A1, A2, B and C (UCYN-A1, UCYN-A2, UCYN-B, and UCYN-C), the heterocystous cyanobacteria symbiotic with diatoms (Het-1, Het-2, and Het-3), the γ-proteobacterium (γ-24774A11), and the α-proteobacterium (α-MH144511). The primer and probe sequences for the TaqMan qPCR assay were summarized in Table S3. The probes were 5'-labeled with the fluorescent reporter FAM (6-carboxy fluorescein) and 3'-labeled with TAMRA (6-carboxytetramethylrhodamine) as a quenching dye. The reference *nifH* sequences used for standard curves were retrieved from PCR clone libraries (i.e., *Trichodesmium* and UCYN-C) or synthesized at GeneWiz Biotechnology Co., Ltd. (Suzhou, China) (Table S2). The cross-reactivity of the primer-probe set was checked using serial dilutions of plasmids containing inserts matching all the other primer and probe sets. Triplicate qPCRs were run for each DNA sample and the standards on a CFX96 real-time system (Bio-Rad, USA). To each reaction, 10–15 ng plasmid standards or environmental DNA were added. The thermocycling conditions were 50°C for 2 min, 95°C for 2 min, and 45 cycles (95°C for 15 s followed by 60°C for 1 min). Negative controls without templates were also included to check contamination. A standard curve was made using serial dilutions of quantified (from $10^1$ to $10^7$ *nifH* gene copies per reaction), linearized plasmid standards for each run. When the standard clone was diluted 10 times, the corresponding Ct value increased by about 3.3–3.4 units, indicating that the PCR amplification efficiencies among all replicates were between 90% and 100% ($R^2 > 0.99$, Fig. S1).

**2.6 Next-generation sequencing of all-inclusive *nifH* phylotypes**

Nested PCR was performed to amplify *nifH* genes from DNA samples using the nested, degenerate *nifH* primers (Zehr et al., 1998) on a MyCycler thermocycler (Bio-Rad, USA). The first round of PCR contained 0.25 μL of Ex Taq (Takara Bio Inc., Japan), 2 μL of the *nifH* primers *nifH*3 (5'-ATRTTRTTNGCNGCRTA-3') and *nifH*4 (5'-TTYTAYGGNAARGGNGG-3') (10 μM each), 5–20 ng of environmental DNA as a template, with a final reaction volume of 50 μL filled with double-distilled H$_2$O. The reaction program consisted of an initial 5 min of denaturation at 95°C, then 31 cycles of 1 min at 95°C, 1 min at 57°C, and 1 min at 72°C, and a final elongation step of 10 min at 72°C. The second round of PCR had the same mixture components and thermocycling conditions (except the annealing temperature of 54°C) but included 2 μL of PCR product from the first reaction and primers *nifH*1 (5'-TGYGAYCCNAARGCNGA-3') and *nifH*2 (5'-TTYTAYGGNAARGGNGG-3') (Zehr and Turner, 2001). The PCR products were visualized by gel electrophoresis and purified using an agarose gel DNA purification kit (Takara Bio Inc., Japan). The purified PCR products were pooled in equimolar amounts and paired-end sequenced (2×300) on an Illumina MiSeq next-generation sequencing (NGS) platform (Illumina, San Diego, USA) according to the standard protocols by Majorbio Bio-pharm Technology

Co. Ltd. (Shanghai, China). The raw sequencing data for the *nifH* gene have been deposited into the NCBI Sequence Read Archive (SRA) database with the accession number PRJNA877297. Raw fastq files were quality-filtered by Trimmomatic and merged by FLASH with the following criteria: i) The reads were truncated at any site receiving an average quality score <20 over a 50 bp sliding window; ii) Sequences having ≥ 10 bp overlap but ≤ 2 bp mismatch were merged (Zhang et al., 2023); iii) Sequences of each sample were separated according to barcodes (exactly matching) and primers (allowing 2 nucleotide mismatching) and reads containing ambiguous bases were removed. Operational taxonomic units (OTUs) were clustered with a 97% similarity cutoff using UPARSE (version 7.1) (http://drive5.com/uparse/) with a novel 'greedy' algorithm (Quince et al., 2011) that performs chimera filtering and OTU clustering simultaneously. OTUs were annotated down to genus level using a formatted *nifH* gene database (https://github.com/moyn413/nifHdada2) that was updated in 2023 according to NCBI and Zehr Lab (https://www.jzehrlab.com/*nifH*).

### 2.7 Determination of abundant and rare taxa

The abundant and rare OTUs were defined based on the relative abundance thresholds of 0.01% and 1%, respectively, as recommended in recent publications (e.g., Dai et al., 2016). The OTUs were classified into six categories: (1) OTUs with abundance >1% in all samples (n=10) were classified as abundant taxa (AT); (2) OTUs with abundance <0.01% in all samples were classified as rare taxa (RT); (3) OTUs with abundance between 0.01 and 1% in all samples were classified as moderate taxa (MT); (4) OTUs with abundance below 1% in all samples and <0.01% in some samples (n<10) were classified as conditionally rare taxa (CRT); (5) OTUs with abundance ≥0.01% in all samples and ≥1% in some samples were classified as conditionally abundant taxa (CAT); (6) OTUs with abundance between 0.01% and 1% were classified as conditionally rare and abundant taxa (CRAT). To avoid confusion with previous definition (Chen et al., 2017), the dominant taxa in the categories AT, CAT and CRAT were pooled and analyzed together as 'AT*' in this study.

### 2.8 Correlation of microbial communities with environmental factors and geographical distance

Mantel's test was used to further reveal the correlations between the diazotrophic community dissimilarity, spatial variables, and environmental factors. The environmental factors are summarized in Table 1. All environmental parameters, except pH, were log (X+1)-transformed to improve homoscedasticity and normality for multivariate statistical analyses and calculation of the Euclidean distances between samples. A set of spatial variables based on the longitude and latitude coordinates of each sampling station were calculated following the approach of the principal coordinates of neighbor matrices (PCNMs) analysis (Borcard and Legendre, 2002). To avoid collinearity among factors, explanatory environmental factors with the highest variance inflation factor (VIF) were eliminated until all VIF values were lower than 20 (Blanchet et al., 2008; Chen et al., 2019). The adjusted $R^2$ was calculated to correlate KIs to distinct diazaotrophs. Redundancy analysis using forward selection identified a minimal subset of environmental and/or spatial variables that explained significant proportions of the variations in the community data ($P < 0.05$), as determined through Monte Carlo permutation test. In addition, diazotrophic compositions were Hellinger-transformed prior to the analyses to provide unbiased estimates of the variation partitioning analysis (VPA) (Legendre & Gallagher, 2001).

### 2.9 Diazotrophic community assembly modeling

To assess the role of KI in shaping diazotrophic community assembly, the occurrence frequency of diazotrophic taxa across the wider metacommunity was fitted to the neutral community model (NCM), a theoretical framework used to explore the potential impact of stochastic processes (Sloan et al., 2007). Calculation of 95% confidence interval was done using 1,000 bootstrap replicates, and the overall fit to the NCM wasindicated by the parameter $R^2$. All data analyses were processed in R (version 4.1.3)

## 3 Results

### 3.1 Hydrographic conditions varied with the intensity of KI

Using the isopycnal mixing model, we integrated all the hydrographic data collected during the cruises in 2017 and 2018 and spotted each corresponding water parcel based on the potential density-temperature-salinity anomalies modeled from the mixing of the Kuroshio and nSCS water masses (Fig. 2a). The sea surface temperature (SST) and salinity for the nSCS were in the lower range (29.20–30.24°C and 33.45–33.81PSU) relative to those for the Kuroshio (29.95–30.81°C and 34.45–34.59 PSU) (Fig. 2a and Table 1). As a result, for the water masses between the endmembers, those with more Kuroshio mixing had higher temperature

and salinity, whereas those with more nSCS mixing had lower temperature and salinity. We further examined this pattern by plotting the depths and temperatures of the isopycnal layer of water masses along a potential density anomaly ($\sigma_\theta$) layer ($\sigma_\theta = 23$ kg m$^{-3}$), and a comparison of the profiles between the two cruises showed that the seawater temperatures were higher and water depths were deeper near the Luzon Strait in 2017 than in 2018, suggesting an intrusion of Kuroshio deeper into the nSCS in 2017 and a gradual withdrawal in 2018 (Figs. 2b-2e). The $I_{NFR}$ in the KC (129.42 ± 100.86 μmol N m$^{-2}$ d$^{-1}$) was higher than that in the

nSCS (97.85 ± 6.98 μmol N m$^{-2}$ d$^{-1}$), whereas $S_{NFR}$ in the nSCS (1.88 ± 0.85 nmol N L$^{-1}$ d$^{-1}$) was higher than that in the KC (0.60 ± 0.25 nmol N m$^{-2}$ d$^{-1}$) as observed along the transect across the Luzon strait (stn1 to stn4). However, the surface primary productivity was found to be similar between the nSCS (0.37 ± 0.12 μmol C L$^{-1}$ d$^{-1}$) and the KC (0.21 ± 0.11 μmol C L$^{-1}$ d$^{-1}$) (Table 1).

### 3.2 The abundance of diazotrophs based on qPCR and NGS

The result of DNA-based quantification of *nifH* gene copies with qPCR showed variations in the abundance of the ten targeted diazotrophic groups between the cruises in 2017 and 2018. *Trichodesmium* and UCYN-B were the most abundant diazotrophs, reaching up to $3.78 \times 10^6$ and $1.10 \times 10^7$ *nifH* gene copies L$^{-1}$, respectively (Table S4). As seen along the transect from Luzon strait to the SCS basin, the *nifH* gene abundance of *Trichodesmium* gradually decreased whereas that of UCYN-B increased (Fig. 3), suggesting that *Trichodesmium* and UCYN-B are the dominant indicator species of Kuroshio and SCS, respectively. Additionally,

the abundance of α-MH144511 and γ-2477A11was much higher (up to $5.49 \times 10^5$ and $5.18 \times 10^4$ *nifH* gene copies L$^{-1}$, respectively) in 2017 when KI was strong (Fig. 3a), while that of UCYN-C was relatively higher in 2018 when KI was weak (Fig. 3b). Moreover, the relative abundance of UCYN-B was higher in 2018 than in 2017, especially at stations close to the SCS gyre (e.g., accounting for 93% of all the targeted diazotrophs at stn9), and the *nifH* gene copies of UCYN-B was higher at night, whereas that of *Trichodesmium* was higher during the day (Table S4). The abundances of other targeted diazotrophs (UCYN-A1, UCYN-A2, Het-

1, Het-2 and Het-3) were mostly less than $1.00 \times 10^4$ *nifH* gene copies L$^{-1}$ in the surveyed years (Table S4).

For DNA-based evaluation of *nifH* amplicons with NGS, the majority (>85%) of the *nifH* amplicons belonged to cyanobacteria and γ-proteobacteria (Table S6). There was considerable station to station variability but across all stations cyanobacteria and γ-proteobacteria were at approximately equal abundances in 2017. In contrast, a higher proportion of cyanobacteria (64%) was detected in 2018 compared to γ-proteobacteria (23%). Overall, *Trichodesmium* and UCYN were

250 approximately 58% and 42%, respectively, of cyanobacterial abundances despite station to station variability. The UCYN-C (22%) and UCYN-B (18%) were the domiant UCYN sublineages in 2017 and 2018, respectively (Table S6).

### 3.3 Correlation between $R_{K\_100}$, diazotrophs, and environmental factors

The relationships between *nifH* gene copies and averaged Kuroshio fractions in the upper 100 m ($R_{K\_100}$, %) showed that *Trichodesmium* was positively correlated with KI ($R^2_{adj}$ = 0.857) whereas UCYN-B was negatively correlated with KI ($R^2_{adj}$ = 0.903) (Fig. 4). The *nifH* NGS showed similar trends, with *Trichodesmium* and UCYN-B being abundant at stations subject to strong and weak KI, respectively, whereas other diazotrophs, including Actinobacteria, Archaea, and Firmicutes, were abundant at stations with a moderate degree of KI (Fig. 5 and Table S5). KI leads to the partition of environmental parameters into two clusters: the depth integrated nitrogen fixation rate ($I_{NFR}$), sea surface temperature (SST), sea surface salinity (SSS), depth of chlorophyll maximum (DCM), and mixed layer depth (MLD) tended to associate with strong KI, whereas the depth-integrated dissolved inorganic phosphorus ($I_{DIP}$), dissolved inorganic nitrogen ($I_{DIN}$), primary production ($I_{PP}$), surface nitrogen fixation rate ($S_{NFR}$), surface primary production ($S_{PP}$), and the depth of nitracline (Nit) with weak KI (Fig. 5).Such variations in environmental parameters mainly affected four diazotrophic groups. Specifically, *Trichodesmium* was distributed at the stations with deep MLD and high SSS and SST, which are characteristics of KI. UCYN-B was distributed at stations least affected by KI and with relatively high nutrient availability. UCYN-C and γ-proteobacteria were mainly distributed at stations moderately affected by the KI (Fig. 6).

### 3.4 Overall fit of diazotrophic community assembly to NCM

The result of Mantel's test showed that environmental distance was significantly more important than the geographical distance in shaping diazotrophic subcommunity structure (Fig. 7). The correlation between diazotrophs and environmental factors (temperature, salinity, MLD, and Chl *a*) was much stronger in 2017 than in 2018, and the effects on all (ALL), abundant (AT), and conditionally abundant taxa (CAT) were consistent in 2017, whereas in 2018 the effects on ALL and AT were consistent (Fig. 7 and Tables S7, S8), suggesting that AT dominated in the survey stations whereas the role of RT was relatively small. The result of VPA further confirmed that the variation in diazotrophic community structure explained by environmental factors was higher in 2017 than in 2018 (0.33 vs. 0.22), whereas the influence of spatial factors was limited and roughly similar between the two years (0.23 vs. 0.26) (Figs. 8a and 8b). This result is also consistent with the NCM, showing that a larger fraction of OTUs fitted to the model in 2017 than in 2018 (Adjusted $R^2$ = 0.676 in 2017 vs. Adjusted $R^2$ = 0.473 in 2018) (Figs. 8c and 8d).

### 4 Discussion

#### 4.1 Pattern of diazotroph distribution in the Kuroshio and nSCS

In this study, by quantifying the degree of KI at each sampling station with the isopycnal mixing model (Du et al., 2013), we were able to compare diazotrophic community structures in the Kuroshio, nSCS, and mixed waters of the two, as well as to explore the pattern in changes of diazotrophic community structure associated with Kuroshio succession on a temporal scale. Overall, our measurements of *nifH* gene copies with qPCR showed that the relative abundance of different diazotrophic phylotypes was highly variable among stations in 2018 but was more uniform in 2017 (Fig. 3), which is likely a result of enhanced upper mixing due to stronger KI in 2017 (Fig. 2). Specifically, *Trichodesmium* and UCYN-B were the two most abundant diazotrophic taxa that dominated the Kuroshio and nSCS, respectively (Fig. 3). Distinct patterns in the distribution of *Trichodesmium* and UCYN-B in the Kuroshio and nSCS accessed from DNA-based qPCR have been reported (Shiozaki et al., 2014). The high abundance of *Trichodesmium* was previously reported in the East China Sea affected by KI during summer (Jiang et al., 2023). This may be attributed to *Trichodesmium*'s higher temperature preference and the coastal input of iron when Kuroshio flows past some islands (Cheung et al., 2017). In contrast, the elevated abundance of UCYN-B in the nSCS may be due to its high iron use efficiency (Sato

et al., 2010), particularly in the iron-depleted nSCS (Shiozaki et al., 2017). Additionally, the relatively lower abundance of UCYN-B in Kuroshio may be due to the greater genetic resources of *Trichodesmium*, which allows it to outcompete UCYN-B for phosphorus (P) (Dyhrman et al., 2006). *Trichodesmium* with gas vacuoles can migrate vertically to the deeper waters to optimize the uptake, utilization, and even storage of P, fulfilling the growth requirement (Sohm et al., 2011; Benavides et al., 2022). It is worth noting that, since we did not observe colonial *Trichodesmium* in our sampling stations, prefiltration using a 100-μm pore-size nylon mesh was not likely to underestimate *Trichodesmium* abundance. While not as abundant as *Trichodesmium* and UCYN-B, the two closely related UCYN-A sublineages, UCYN-A1 and UCYN-A2, were present at all the stations surveyed in both 2017 and 2018 (Fig. 3). The distribution and abundance patterns of UCYN-A1 and UCYN-A2 were similar between the Kuroshio and nSCS, and between the two cruises, which seems at odds with the previous view that these two UCYN-A sublineages may have distinct niches (Thompson et al., 2014). The reason is unknown, perhaps more extensive sampling towards the coastal regions may provide clues as it has been reported that coastal environments favor the proliferation of UCYN-A2 (Gradoville et al., 2021). Furthermore, the abundant UCYN-A sustained in the upstream Kuroshio could also be a possible reason why UCYN-A can be readily detected in higher-latitude regions than other cyanobacterial diazotrophs (Martínez-Pérez et al., 2016; Henke et al., 2018).

Our data show that UCYN-C was among the least abundant diazotrophs in 2017 when KI was strong, but its relative abundance was 1–2 orders of magnitude higher in 2018 when KI was weak (Fig. 3). UCYN-C has rarely been reported in other regions and was among the poorly understood diazotrophs. Many *nifH* sequences of UCYN-C have been recovered from downstream Kuroshio in the Tokara Strait of Japan and the high abundance of UCYN-C is considered one striking feature of Kuroshio (Cheung et al., 2017). An alternative explanation could be that the UCYN-C in downstream Kuroshio may originate from the nSCS, as inferred from the negative correlation of UCYN-C abundance with KI in our interannual comparison. In fact, the dominance of UCYN-C in phosphate-depleted conditions has been reported (Turk-Kubo et al., 2018), suggesting that UCYN-C may adapt well to the nutrient-limited conditions of the nSCS (Wu et al., 2003). In contrast to UCYN-C, the abundance of α- and γ-proteobacteria was much higher in 2017 than in 2018, suggesting that these NCDs could potentially be transported by Kuroshio to the nSCS. The cross-station similar abundance of α- and γ-proteobacteria (Fig. 3) also confirms that these two groups occupy similar ecological niches (Turk-Kubo et al., 2022).

**4.2 Contribution of KI to nutrient inventory in the nSCS**

The vertical pattern of water exchange between the nSCS and the WPO through the Luzon strait mimics a sandwich structure, with waters flowing into the nSCS at depths of 0–400 m and ~1,500–3,000 m and returning to the WPO at intermediate depths of ~500–1,500 m (Nan et al., 2015). Isopycnal modeling of water mass mixing showed that the nutrient inventory in the upper 100 m of the nSCS was overall negatively correlated with the Kuroshio water fraction, suggesting a dominant dilution effect of KI on the surface nutrient budget of nSCS, which typically features the oligotrophic Kuroshio waters (Du et al., 2013). The high salinity of Kuroshio waters also contributes to the overall negative correlation between the nutrient inventory and Kuroshio water fraction due to a downward mixing of the saline waters to deeper isopycnal surfaces that prevents nutrient upwelling from the subsurface (i.e., deeper nutricline) (Wu et al., 2015). The isopycnal modeling result is consistent with our *in situ* measurements showing that high nutrient availability was associated with the stations least affected by KI (Figs. 5 and 6). The dilution effect of KI on surface nutrient budget in nSCS could provide favorable ecological niches for diazotrophs, limiting other phytoplankton which can not conduct $N_2$ fixation (Zehr & Capone 2020). This might partially elucidate why KI has the potential to enhance $N_2$ fixation rates in the nSCS. The dilution effect was even more pronounced under the El Niño'-driven, intensified KI (Ding et al., 2022). It might be argued that nutrient supplies from the depths through diapycnal mixing could counterbalance the dilution effect of KI. However, this is most likely to occur during the northeast monsoon in winter when diapycnal mixing is intensified by strong wind stress curl

and internal wave shoaling. Since our cruises were carried out during the intermonsoon season and the strong stratification in the upper mixed layer diminishes the influx of nutrient-rich subsurface waters through diapycnal mixing, the observed nutrient dynamics are thus more affected by isopycnal mixing.

Besides the abiotic mixing, biogeochemical processes also contribute significantly to the nutrient inventory in the upper ocean. One such process is $N_2$ fixation. The Kuroshio, characterized with high temperature, high salinity, and low nutrient concentrations, provides a relatively stable environment in strengthening $N_2$ fixation by delivering oceanic diazotrophs to the photic zone (Tang et al., 2000). $N_2$ fixation was estimated to be ~20 mmol N m$^{-2}$ yr$^{-1}$ and account for less than 10% of the present nutrient inventory in the nSCS (~250 mmol N m$^{-2}$ yr$^{-1}$) (Kao et al., 2012), a fraction that may reflect different nutrient dynamics between the KC

and nSCS. While the unicellular diazotrophs were reported to contribute more than 50% of $N_2$ fixation in the nSCS (Chen et al., 2014; Wu et al., 2018), *Trichodesmium* was primarily responsible for the enhanced $N_2$ fixation and primary production in the KC (Lu et al., 2019). Similar to previous findings (Chen et al., 2014), the $I_{NFR}$ in this study was higher in the KC than at nSCS (Table 1). In contrast, relatively lower $S_{NFR}$ was observed in the KC, possibly due to the deeper nitracline inhibiting upward transport of nutrients (Chen et al., 2008). This distinct patterns in surface and deep-water nitrogen fixation rates might be explained by the

niches adaptaton of different diazotrophs, whereby KI-transported *Trichodesmium* typically dominates shallow warm water (< 50 m) (Jiang et al., 2015; Jiang et al., 2019), whereas UCYN-B prefers to live in lower temperature and deeper water (50–75 m) (Moisander et al. 2010). The KI-induced frontal zone mixing in the upper water column also causes redistribution of the diazotrophs and reallocation of the nutrients (Jiang et al., 2019; Kuo et al., 2020). In addition to $N_2$ fixation, previous studies also showed that KI introduces external DON into the nSCS and thereby stimulates ammonia oxidation, further complicating the conventional

concept of N-based new production (Xu et al., 2018).

**4.3 Impact of Kuroshio as a stochastic process and potential implications to climate change response**

The mechanisms by which KI drives the biogeography of marine diazotrophs are complicated and remain a central issue to microbial ecology. Our result based on *nifH* qPCR and NGS shows a lineage-specific distribution of diazotrophs in response to environmental changes associated with KI. Differences in the types and abundances of diazotrophs may arise through selection-

driven (deterministic) and/or non-selection driven (stochastic) processes. Deterministic processes may drive differences between communities through species sorting in response to local environmental conditions, while stochastic processes may generate variation through a combination of other assembly processes including dispersal limitation, community drift and speciation (Hughes et al., 2008; Hanson et al., 2012). These stochastic processes—which we define here as 'neutral' processes—are considered in ecological neutral theories and are predicted to produce variation in community structure through space without

needing to invoke the actions of selection. Specifically, *Trichodesmium* was predominated at the stations with high SSS and SST and deep MLD, which feature strong KI, whereas UCYN-B was distributed at stations least affected by KI and with relatively high nutrient availability (Figs. 5 and 6). Furthermore, in contrary to many studies investigating microbial community with no distinction between the abundant and rare ecotypes, we analyzed whether environmental and spatial  factors have differing impacts on diazotrophic subcommunities, including ALL, dominant (AT* = AT+CAT+CRAT), CRT, and RT. We found that environmental

factors (e.g., temperature, salinity, MLD, and Chl *a*) were highly correlated with the overall diazotrophic community and the AT* subcommunity, and this correlation appeared to be much stronger in 2017 than in 2018, but no environmental variables were significantly correlated with the RT subcommunity (Fig. 7). This result suggests that the AT* and RT subcommunities may response differently to environmental variables, or that RT may not bet constrained by environmental variables possibly due to their low growth rate, low competition potential, and narrow resource range (Pedros-Ali., 2006; Reveillaud et al., 2014).Among the

environmental parameters correlated with KI, temperature is a major factor controlling the distribution of diazotrophs. For example,

*Trichodesmium* is distributed in surface waters between 20°C and 30°C and thrives at 25°C or warmer temperatures (Capone, 1997). In contrast, unicellular diazotrophs are often found in deeper and colder waters although the optimal temperature for UCYN-A and UCYN-B were different (Moisander et al., 2010). Salinity is another important environmental factor affecting the osmoregulation, metabolism, and community composition of diazotrophs (Zehr and Turner, 2001; Reeder et al., 2022). Salinity was also reported as the major determinant of abundant and rare bacterial subcommunities in freshwater ecosystem (Kumar et al., 2018). Taken together, the above findings highlight the role of dominant diazotrophic lineages (e.g., *Trichodesmium* and UCYN-B) in shaping the overall diazotrophic community structure associated with KI, whereas the role of RT is relatively small.

Both Mantel's test and VPA reveal that the overall diazotrophic community structure was more significantly affected by environmental factors in 2017 when KI was strong than in 2018 when KI was weak, whereas the influence of spatial factors was limited and roughly similar between the two years (largely due to sampling within the small latitudinal range from 16°N to 22°N) (Figs. 7, 8a and 8b). This result is also consistent with neutral model simulation showing that the diazotrophic community was more likely affected by KI primarily as a stochastic process (Figs. 8c and 8d). It should be noted, however, that the variation in diazotrophic community structure explained by the spatial and the environmental factors associated with KI is still relatively low (i.e., 64% and 56% of unexplained variation in 2017 and 2018, respectively; Figs. 8a and 8b). The unexplained variation may be caused by other biotic interactions such as competition from non-diazotrophic microorganisms (Turk-Kubo et al., 2018) and microzooplankton grazing (Deng et al., 2020), or other abiotic parameters that were not measured in the study. The distribution of endemic diazotrophic populations may be under the control of diverse physical processes governing upper mixing in the nSCS, such as fluctuations in Kuroshio current (Shiozaki et al., 2015a), surface wind curl, tides, typhoons (Cheung et al., 2020), mesoscale eddies, upwelling and abyssal circulation (Kuo et al., 2020). For instance, high-resolution climate modeling predicts that the velocity of Kuroshio can reach 0.3 m s$^{-1}$ due to global warming, which may influence the northward heat transport at the basin scale (Sakamoto et al., 2005). Since N$_2$ fixation by marine diazotrophs has been proposed as one of the potential negative feedback mechanisms corresponding to ocean warming (Sohm et al., 2011), Kuroshio may transport diazotrophs in the upstream warmer regions including SCS northward to higher latitudes, resulting in a wider distribution of N$_2$ fixation in the global ocean. It is, therefore, worthy of efforts to explore the effects of KI on the biogeography of diazotrophic communities on a large scale, for example, through extensive sampling across wider geographic ranges, collection and monitoring of environmental factors with more sophisticated devices such as the biogeochemical Argo floats (Wang et al., 2020a), and performing manipulative experiments indoors and outdoors. Nevertheless, given that KIs are projected to intensify in a future warming ocean (Chen et al., 2019), our analysis provides a case study in evaluating the role of KI in shaping diazotrophic community structure primarily as a stochastic process, which has implications for the redistribution of diazotrophs and nutrient budget at local and regional scales.

# 5 Conclusions

In this study, we surveyed the diversity and abundance of diazotrophic phylotypes in the nSCS in two consecutive years based on molecular approaches targeting the nitrogenase gene *nifH* and associated the patterns in changes of diazotrophic community structure in the upper 100 m with the degrees of KI quantified using the isopycnal mixing model. Our study reveals a lineage-specific niche adaptation to environmental changes associated with KI in the overall diazotrophic community structure in the nSCS. Four diazotrophic groups, *Trichodesmium*, UCYN-B, UCYN-C, and γ-proteobacteria, appear to be highly correlated with KI leading to variations in a range of physicochemical parameters. KI has a dominant dilution effect on the nutrient inventory on one hand, it also causes redistribution of the diazotrophic taxa and reallocation of the nutrients on the other hand. Neutral model simulation suggests that the diazotrophic community is more likely affected by KI primarily as a stochastic process. As KIs are

projected to intensify in a future warming ocean, Kuroshio may potentially cause a wider distribution of diazotrophs at high latitudes.

**Data availability**

All data needed to evaluate the conclusions in the paper are present in the paper and/or the Supplement. Additional data associated with the paper are available from the corresponding authors upon request.

**Author contributions**

T.S. conceived and designed the study. W.L. and R.D. participated in the expedition cruises and collected the samples. H.Z. and W.L. performed laboratory experiments. G.M. and M.C. contributed to the reagents, materials, and analysis tools. H.Z., W.L., and T.S. analyzed the data. H.Z. and T.S. drafted the manuscript. All authors read and approved the final version of the manuscript.

**Competing interests**

The authors declare that they have no conflict of interest.

**Acknowledgements**

The authors are grateful to the captain and crew of the R/Vs 'Dong Fang Hong 2' and 'Tan Kah Kee' for logistics at sea and help with collection of the hydrographic data. Special thanks also go to Dalin Shi for providing some of the nutrients and $N_2$ fixation rates data, Kangkai Li and Linggang Zheng for assistance with sampling, and Zuozhu Wen, Mingming Chen, and Weidong Chen for discussion with data analysis. The authors also thank the reviewers for their insightful comments that help improve the clarity of the manuscript. This study was supported by the National Natural Science Foundation of China (grant numbers 41676092 and 42076152 to T.S.).

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

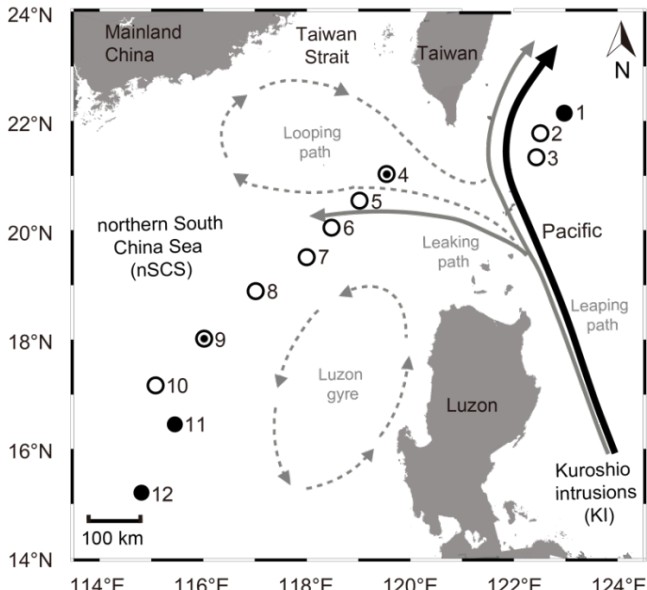

Figure 1. Sampling stations in the northern South China Sea (nSCS ) during the cruises in 2017 and 2018. Solid and empty circles represent the sampling stations in 2017 and 2018, respectively, whereas a combination of solid and empty circles represents the sampling stations in both years (i.e., stn4 and stn9). Major patterns of Kuroshio intrusions into the nSCS, i.e., the looping path (grey dashed), the leaking path (grey solid), and the Kuroshio leaping across the Luzon Strait with no penetration (black) are indicated. The leaking path represents the Kuroshio bifurcating with one branch entering the nSCS and the other taking the leaping path. The looping path represents the Kuroshio entering the nSCS, retroflecting, and then bending eastward back into the western Pacific. The Luzon gyre (long grey dashed) is a typically unstable offshoot near the Luzon Strait when Kuroshio passes by the Luzon Strait. The map was constructed using Ocean Data View 5.1.0.

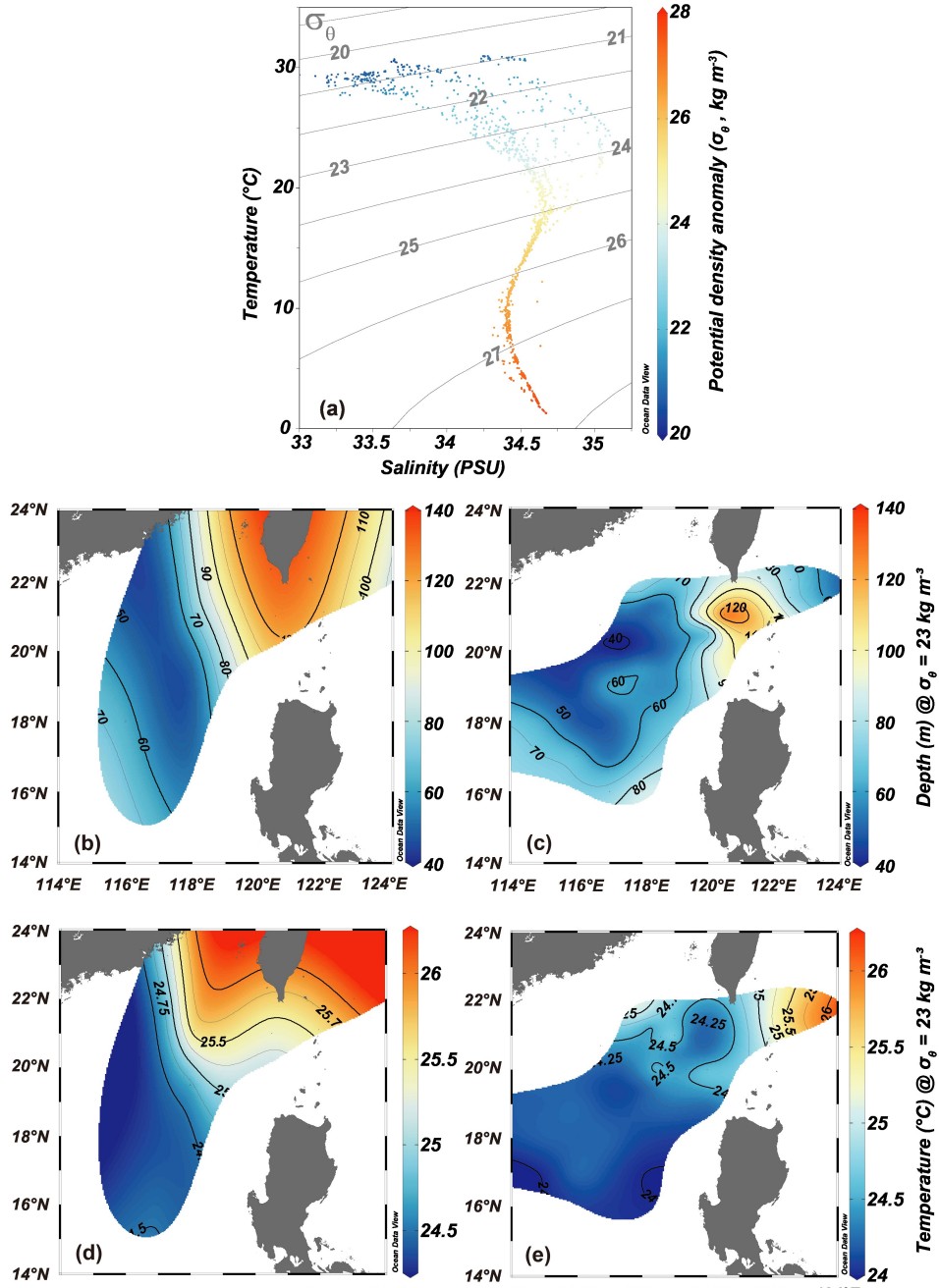

Figure 2. Potential temperatures, salinities and density anomalies modeled for the cruises in 2017 and 2018. (a) Plot of $\theta$-$S$ showing potential temperatures ($\theta$, °C) and salinities (S, PSU) of water parcels resulting from mixing of the Kuroshio and SCS water masses. Potential density anomalies ($\sigma_\theta$, kg m$^{-3}$), shown in grey lines, are imposed on the $\theta$-$S$ plot. (b–e) Profiles of depths (b, c) and temperatures (d, e) of isopycnal surface along the $\sigma_\theta$ of 23 kg m$^{-3}$ of water masses depicted for the cruises in 2017 (b, d) and 2018 (c, e). The data points in (a) are the real data used to create the contour plots in panels b–e.

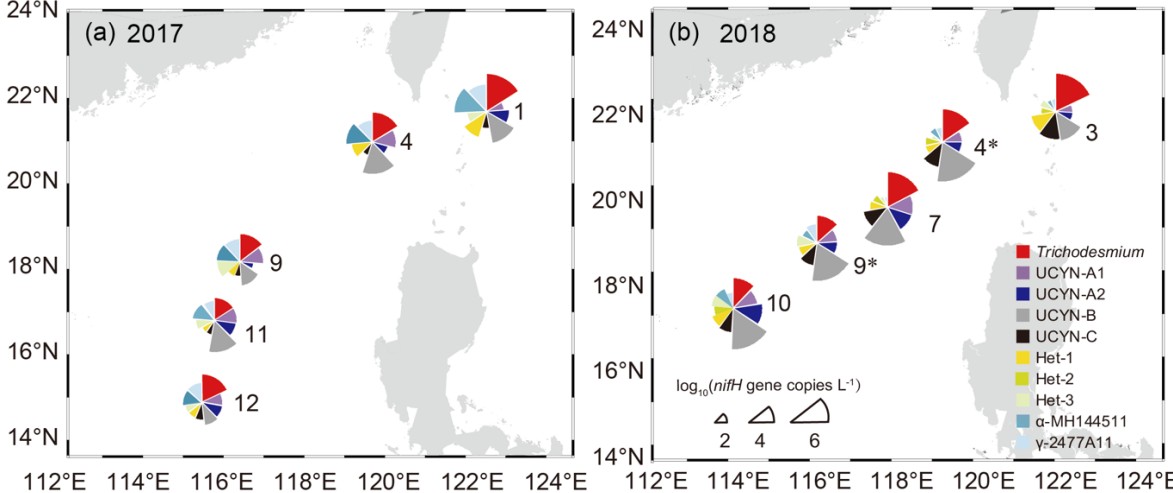

Figure 3. Surface daytime abundance of the ten major diazotrophic groups determined based on TaqMan qPCR assay of the *nifH* gene copies in samples collected in 2017 (a) and 2018 (b). Sector radius represents log-transformed absolute abundance of the *nifH* gene copies, with the angle indicating the relative proportion of each group. For the stations sampled in both years, the station number in 2018 are indicated with an asterisk sign (*).

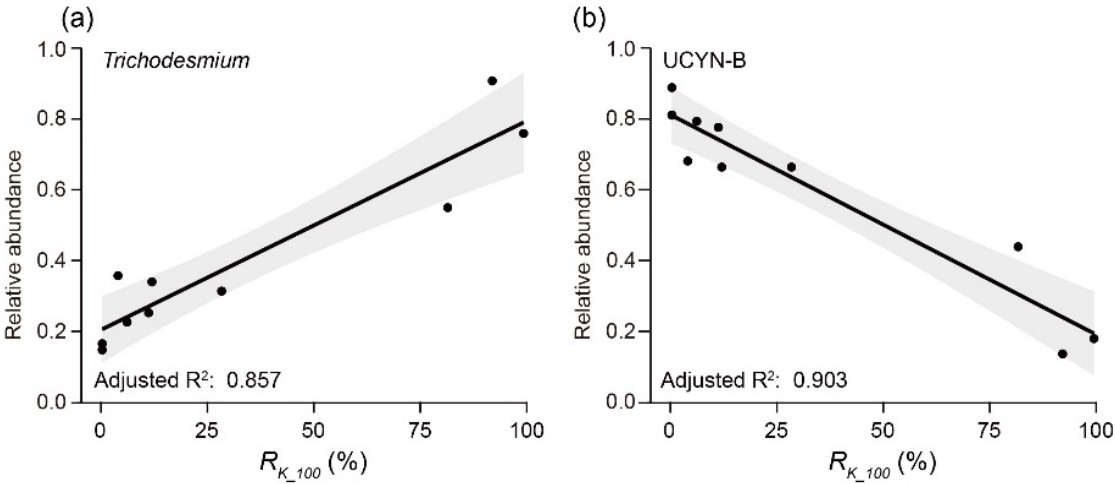

Figure 4. Correlation between KI and relative abundance of the two dominant diazotrophic groups based on *nifH* gene copies. *Trichodesmium* (a) and UCYN-B (b) exhibited contrast patterns in their relationships with the averaged Kuroshio fractions in the upper 100 m ($R_{K\_100}$, %). The values of adjusted $R^2$ represent the degree to which the relative abundance of each group in explaining the $R_{K\_100}$. Linear regressions are indicated with a trendline with 95% confidence interval.

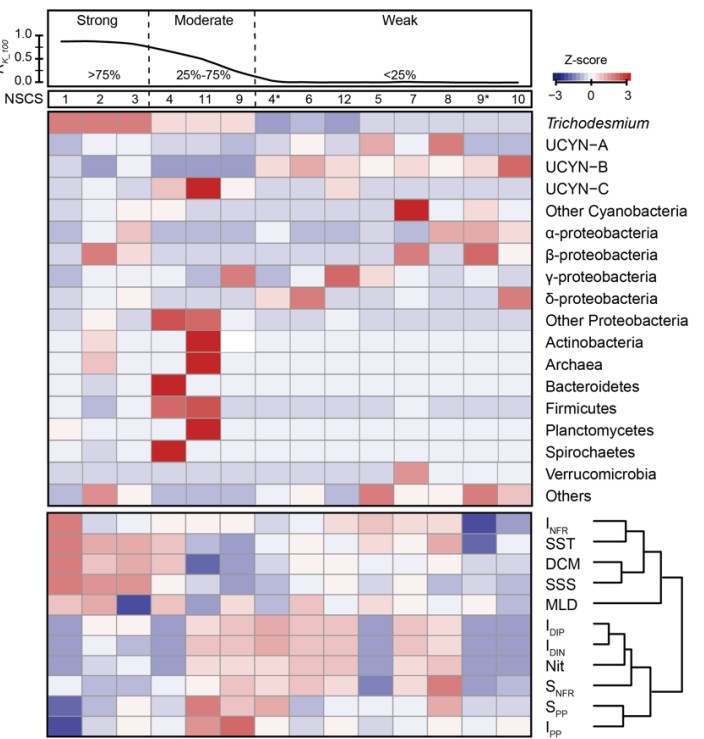

730

Figure 5. Correlations of KI with diazotrophic community compositions and environmental factors. The averaged Kuroshio fractions in the upper 100 m ($R_{K\_100}$, %) at each sampling station are plotted in the upper panel, indicating the degree of KI (strong, >75%; moderate, 25%–75%; weak, <25%). The value of Z-score between $R_{K\_100}$ and the relative abundance of specific diazotrophic groups (middle panel) or environmental factors (lower panel) at each sampling station are also plotted. Stations sampled in both years are indicated with an asterisk sign (*). The degrees of KI are color-coded based on $R_{K\_100}$ values ($R_{K\_100}$>75%, strong intrusion; 25%≤$R_{K\_100}$≤75%, moderate intrusion; $R_{K\_100}$<25%, weak intrusion). $I_{NFR}$, depth-integrated N2 fixation rate; SST, surface seawater temperature; DCM, depth of chlorophyll maximum; SSS, sea surface salinity; MLD, mixed layer depth; $I_{DIP}$, depth-integrated dissolved inorganic phosphorus; $I_{DIN}$, depth-integrated dissolved inorganic nitrogen; Nit, nitraicline; $S_{NFR}$, surface nitrogen fixation rate; $S_{PP}$: surface primary production; $I_{PP}$, depth-integrated primary production; The environmental factors used in this study are shown in Table 1.

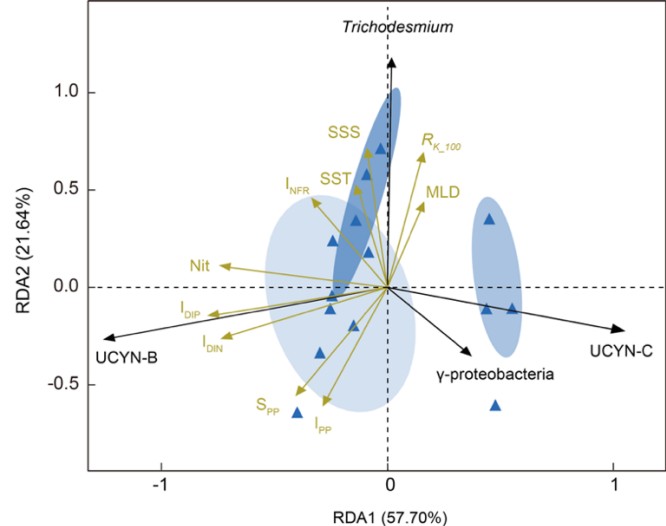

Figure 6. Redundancy analysis shows ordination of the major diazotrophic groups in relation to environmental factors under the influence of KI. Four diazotrophic groups, *Trichodesmium*, UCYN-B, UCYN-C, and γ-proteobacteria, appear to be highly correlated with KI leading to variations in a range of physicochemical parameters. The degrees to which KI potentially affect diazotrophic community members and environmental factors are indicated with arrows in black and yellow, respectively. Sampling stations under the influence of different degrees of KI are color-coded based on $R_{K\_100}$ values and clustered given a 95% confidence threshold ($R_{K\_100}$>75%, strong intrusion, in dark blue ellipse; 25% $\leq R_{K\_100} \leq$75%, moderate intrusion, in moderate blue ellipse; $R_{K\_100}$ <25%, weak intrusion, in light blue ellipse). $I_{NFR}$, depth-integrated $N_2$ fixation rate; SST, surface seawater temperature; SSS, sea surface salinity; MLD, mixed layer depth; $I_{DIP}$, depth-integrated dissolved inorganic phosphorus; $I_{DIN}$, depth-integrated dissolved inorganic nitrogen; Nit, nitraicline; $S_{PP}$: surface primary productivity; $I_{PP}$, depth-integrated primary productivity; $R_{K\_100}$, averaged Kuroshio fractions in the upper 100 m ($R_{K\_100,}$ %); The environmental factors used in this study are shown in Table S4.

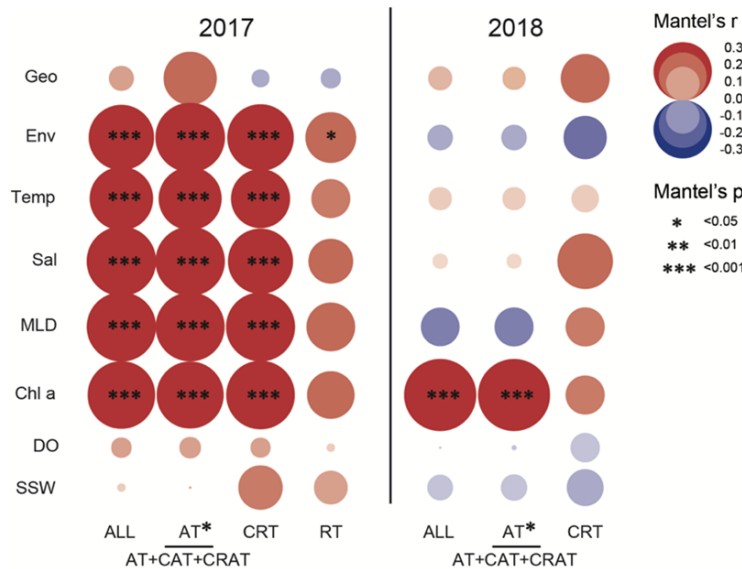

Figure 7. Mantel's test of the effects of spatial and environmental variables on diazotrophic subcommunities. Spearman's coefficients were used to evaluate the effects of geographical and environmental distances, as well as individual environmental parameters on both abundant and rare diazotrophic taxa. The partitioning of these effects was performed with Mantel's test and compared between data collected in 2017 and 2018. Geo, geographical distance; Env, environmental distance; Temp, temperature; Sal, salinity; MLD, mixed layer depth; Chl $a$, Chlorophyll $a$; DO, dissolved oxygen; SSW, sea surface wind; All, all taxa; AT, abundant taxa; CAT, conditionally abundant taxa; CRAT, conditionally rare and abundant taxa; CRT, conditionally rare taxa; RT, rare taxa.

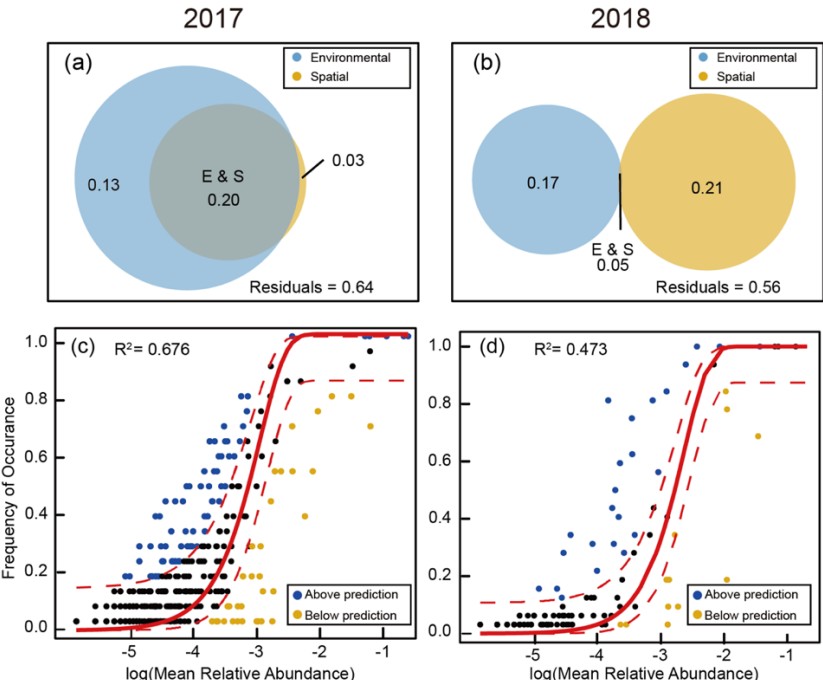

Figure 8. Simulation of KI as a stochastic process in shaping diazotrophic community structure using the NCM. A comparison
between the samples collected in 2017 (a, c) and 2018 (b, d) was conducted. The Venn diagrams (a, b) show the variations in
diazotrophic community structure explained solely by environmental or spatial factors, or by the combination of the two.
Representative OTUs of diazotrophic community assembly that best fit the NCM are shown as black dots surrounding the
regression lines (solid) within a 95% confidence interval (dashed lines) (c, d), whereas those exceeding the fitted model prediction
range are shown as colored dots.

Table 1. Summary of environmental factors detected in this study.

| Year | Station | SST (°C) | SSS (PSU) | MLD (m) | DCM (m) | Nit (m) | $I_{DIN}$ (mmol m$^{-2}$) | $I_{DIP}$ (mmol m$^{-2}$) | $I_{NFR}$ (μmol N m$^{-2}$ d$^{-1}$) | $S_{NFR}$ (nmol N L$^{-1}$ d$^{-1}$) | $I_{PP}$ (mmol C m$^{-2}$ d$^{-1}$) | $S_{PP}$ (μmol C L$^{-1}$ d$^{-1}$) | $R_{K\_100}$ (%) |
|------|---------|-----|-------|----|-----|-----|--------|--------|--------|------|--------|------|--------|
| 2017 | stn1   | 30.81 | 34.59 | 40 | 137 | ND  | ND     | ND     | 271.88 | 0.90 | 4.67   | 0.06 | 100.00 |
| 2017 | stn4*  | 29.95 | 33.85 | 37 | 98  | ND  | ND     | ND     | 86.40  | 0.71 | 21.11  | 0.24 | 82.01  |
| 2017 | stn9*  | 29.71 | 33.60 | 34 | 69  | ND  | ND     | ND     | 129.81 | 0.11 | 18.09  | 0.29 | 11.81  |
| 2017 | stn11  | 27.96 | 33.64 | 32 | 63  | ND  | ND     | ND     | 11.73  | 0.25 | 20.87  | 0.23 | 28.34  |
| 2017 | stn12  | 29.20 | 33.43 | 22 | 63  | ND  | ND     | ND     | 27.27  | 0.39 | 27.7   | 0.26 | 5.89   |
| 2018 | stn3   | 29.17 | 34.51 | 15 | 117 | 131 | 1.00   | 8.00   | 64.38  | 0.48 | 27.76  | 0.32 | 92.57  |
| 2018 | stn4*  | 28.87 | 33.66 | 21 | 44  | ND  | ND     | 16.52  | 77.95  | 0.96 | 90.00  | 0.61 | 11.01  |
| 2018 | stn5   | 28.59 | 33.30 | 36 | 52  | ND  | ND     | 19.41  | 86.82  | 1.57 | 121.16 | 0.43 | 5.83   |
| 2018 | stn6   | 29.35 | 33.46 | 22 | 64  | 26  | 25.00  | 179.50 | 52.19  | 1.17 | 27.11  | 0.52 | 6.23   |
| 2018 | stn7   | 29.50 | 33.78 | 40 | 83  | 50  | 28.00  | 230.11 | 56.49  | 1.75 | 19.39  | 0.20 | 3.75   |
| 2018 | stn8   | 29.20 | 33.84 | 27 | 78  | 50  | 272.23 | 23.38  | 105.90 | 1.38 | 26.89  | 0.31 | 0.44   |
| 2018 | stn9*  | 29.55 | 33.81 | 31 | 71  | 44  | 177.42 | 17.21  | 94.03  | 1.41 | 21.09  | 0.30 | 0.00   |
| 2018 | stn10  | 30.24 | 33.45 | 27 | 82  | 56  | 95.80  | 12.18  | 93.61  | 2.86 | 31.55  | 0.51 | 0.00   |

Note: The same stations sampled in both 2017 and 2018 are marked with an asterisk (*). Abbreviations: SST, sea surface temperature; SSS, sea surface salinity; MLD, mixed layer depth; DCM, depth of chlorophyll maximum; Nit, depth of nitracline; $I_{DIN}$, depth-integrated dissolved inorganic nitrogen; $I_{DIP}$, depth-integrated dissolved inorganic phosphorus; $S_{NFR}$, surface N$_2$ fixation rate; $I_{PP}$, depth-integrated primary production; $S_{PP}$, surface primary production; $R_{K\_100}$, averaged Kuroshio fractions in the upper 100 m; ND, not determined.

775