# Peer review of "Changes in diazotrophic community structure associated with Kuroshio succession in the northern South China Sea"

_Biogeosciences, 2023_

## Author Comment (AC1)

"Changes in diazotrophic community structure associated with Kuroshio succession in the northern South China Sea" by Han Zhang et al.

We have taken all the comments of the Reviewers into account in the revision. Our point-by-point responses are provided below in blue fonts. Please note that all the line numbers mentioned in the response refer to the Marked-up Manuscript.

**General comments:**

**Reviewer #2 (Comments for the Author):**

1. The study explores the impact of interannual Kuroshio intrusions (KIs) on diazotrophic communities in the northern South China Sea. The research identifies a correlation between diazotroph variations and KI-induced changes in environmental factors. Notably, *Trichodesmium* abundance and $N_2$ rate increases in strongly affected stations, while unicellular $N_2$-fixing cyanobacteria (UCYN-B) thrive in less affected stations. UCYN-C and non-cyanobacterial $\gamma$-proteobacteria dominate moderately affected stations. The findings suggest KI succession shapes diazotrophic communities as a stochastic process, potentially influencing region-scale redistribution of diazotrophs and nitrogen budgets in a warming ocean. I find the work excellent and should be considered for acceptance, however some minor revision is needed.

Response:

We thank the Reviewer for the comments. We are thankful for considering this manuscript for acceptance. We hope that our point-by-point response below will address your concern.

2. **Introduction:** As there is some excellent work done with non-cyanobacteria diazotrophs I recommend a deeper introduction into those. As a start, investigate "non-cyanobacterial diazotrophs: global diversity, distribution, ecophysiology, and activity in marine waters" by Kendra A. Turk-Kubo to get an overview.

Response:

We are grateful for the Reviewer's comments. We agree with the Reviewer that there is some excellent work done with non-cyanobacteria diazotrophs, and that non-cyanobacterial diazotrophs could play an important role in global marine $N_2$ fixation. Therefore, we have added further description about non-cyanobacterial diazotrophs in Introduction as below:

Line 52-54: "NCDs have been reported to be ubiquitous in ocean ecosystems and contribute to global marine $N_2$ fixation (Moisander et al, 2014; Chakraborty et al, 2021; Turk-Kubo et al 2022).".

**Method:**

3. Regarding $N_2$ fixation and productivity rate. What elemental analyzer were used (i.e. model).

Response:

We are grateful for the Reviewer's comments. We have added further description about elemental analyzer in Method as below:

Line 136-138: "To estimate the natural and tracer-enriched $^{15}$N and $^{13}$C abundance, the samples were first acid fumed to remove the inorganic carbon and then analyzed using an elemental analyzer coupled to a mass spectrometer (Flash HT 2000-Delta V Plus, Thermo Fisher Scientific)."

4. You used *nifH* ARB database from john Zehr. However, this is from 2017 and has not been updated. I suggest that you use the new nifH database, based on ARB and John zehr but updated in 2023 according to the NCBI. See link https://github.com/moyn413/nifHdada2

Response: We are thankful for the Reviewer's suggestion. We have used the updated *nifH* database to conduct the analysis and got the same results. We have added further description about elemental analyzer in Method as below:

Line 187-189: "OTUs were annotated down to the genus level using a DADA2 formatted nifH gene database (https://github.com/moyn413/nifHdada2) that was updated in 2023 according to NCBI and Zehr Lab (https://www.jzehrlab.com/nifH).".

5. It would be helpful to know if you did qPCR on all stations and if you did sequence of all stations or only some.

Response:

We are thankful for the Reviewer's comments. The DNA samples on all stations (Fig. 1) were used as templates to quantify the *nifH* gene copies using TaqMan qPCR technique targeting 10 major diazotrophic phylotype, and nested PCR was also performed to amplify *nifH* genes sequence from DNA samples on all stations using the nested, degenerate *nifH* primers.

**Discussion:**

6. At section 4.1, I think that the paper could benefit from a deeper discussion about non-cyanobacterial diazotrophs.

Response:

We are thankful for the Reviewer's suggestion. We have added a deeper discussion about non-cyanobacterial diazotrophs as below:

Line 312-315: "In contrast to UCYN-C, the abundance of non-cyanobacterial diazotrophic α- and γ-proteobacteria was much higher in 2017 than in 2018, suggesting that heterotrophic diazotrophs could potentially be transported by Kuroshio to the nSCS. The close abundance of non-cyanobacterial diazotrophic α- and γ-proteobacteria (Fig. 3) indicates that these two groups occupy similar ecological niches (Turk-Kubo *et al.* 2022).".

**Specific comments:**

7. Title: Should it not be "changes in diazotrophic community....."

Response:

We thank the Reviewer for the suggestion, and we have revised the Title as "Changes in diazotrophic community structure associated with Kuroshio succession in the northern South China Sea"

8. In line 18 at the abstract, you write "..non-cyanobacterial gamma-proteobacteria..", you can delete either non-cyanobacteria or gamma-proteobacteria. As it is evident that a gamma-proteobacteria is a NCD.

Response:

Thanks for the suggestion and we have deleted "non-cyanobacterial". in Abstract as below:

Line18-19: "……,whereas UCYN-C and the non-cyanobacterial γ-proteobacteria were prevalent at stations moderately affected by KI.".

9. Line 20 - remove "more".

Response:

Thanks for the suggestion and we have removed it. Please refer to Line 20.

10. Line 20-21 – Rephrase "..in 2017 when KI was strong than in 2018 when KI appeared to have retreated." Or perhaps just say "..when KI was stronger compared to 2018 where KI retreated". If I understood it correctly

Response:

We are thankful for the Reviewer's suggestion. We have revised the description as bellow:

Line 19-21: "Neutral community model further demonstrated that dominant diazotrophic subcommunities were more significantly affected by environmental factors in 2017 when KI was stronger compared to 2018 when KI retreated.".

11. Line 50-52 - Perhaps expand the importance of these new groups? e.g., they have been models and measurement of NCDs contribution to $N_2$ fixation.

Response:

We thank the Reviewer for the comments. We have added the description about the importance of these new groups as below:

Line 50-54: "Particularly, diazotrophs genomes reconstructed from global metagenomic data have extended the PCR-based amplicon surveys, revealing new diatom-diazotroph symbioses (Schvarcz et al., 2022) and new species of non-cyanobacterial diazotrophs (Bombar et al., 2016; Delmont et al., 2018). NCDs have been reported to be ubiquitous in ocean ecosystems and contribute to global marine $N_2$ fixation (Moisander et al, 2014; Chakraborty et al, 2021; Turk-Kubo et al 2022).".

12. Line 180 - were relative abundance of OTU determined from UPARSE? Or how was this determined?

Response:

We thank the Reviewer for the comments. We determined relative abundance of OTU using UPARSE with a 97% similarity cutoff.

Line 180-187: "Raw demultiplexed paired-end reads were processed into Operational taxonomic units (OTUs) were clustered with a 97% similarity cutoff using UPARSE (version 7.1) (http://drive5.com/uparse/) with a novel 'greedy' algorithm that performs chimera filtering and OTU clustering simultaneously. OTUs were annotated down to the genus level using a DADA2 formatted

nifH gene database (https://github.com/moyn413/nifHdada2) that was updated in 2023 according to NCBI and Zehr Lab(https://www.jzehrlab.com/nifH).".

Sequence data were not rarified before analysis as this may lead to a loss of information and our minimum sequence depth of ca. 36519 reads per sample exceeded the depth considered acceptable for comparing microbiome composition between samples. Functions within the package 'phyloseq' (v. 1.34.0; McMurdie and Holmes, 2013) were used to generate a phyloseq object containing the filtered count table, taxonomy table, and contextual sample data. Statistical analyses were performed with compositional data based on the "total sum" relative read abundances.

13. Line 184 - you mention "some samples". What is the threshold? How many samples? below or above e.g. 10?

Response:

We are thankful for the Reviewer's comments. The total number is 10, so the threshold is 10. Here, "some" means the number of samples is below the threshold 10. We have clarified this in the revised manuscript.

Line 193-199: The OTUs were classified into six categories: (1) OTUs with abundance >1% in all samples (n=10) were classified as abundant taxa (AT); (2) OTUs with abundance <0.01% in all samples were classified as rare taxa (RT); (3) OTUs with abundance between 0.01 and 1% in all samples were classified as moderate taxa (MT); (4) OTUs with abundance below 1% in all samples and <0.01% in some samples (n<10) were classified as conditionally rare taxa (CRT); (5) OTUs with abundance ≥0.01% in all samples and ≥1% in some samples were classified as conditionally abundant taxa (CAT); (6) OTUs with abundance between 0.01% and 1% were classified as conditionally rare and abundant taxa (CRAT).".

14. Line 191 - why not transform pH?

Response:

We thank the Reviewer for the comments. The pH data in this study was close to normal distribution. Therefore, the pH data was not transformed.

15. Line 268-269 - I would recommend to include station number, to help the reader.

Response:

We are thankful for the Reviewer's comments. We have added the station number in the manuscript as below:

Line 283-284: "Specifically, the two most abundant diazotrophs were *Trichodesmium* and UCYN-B, which dominated the Kuroshio (stn11) and nSCS (stn4*), respectively (Fig. 3)."

16. Line 275 - Regarding sinking *Trichodesmium*. Check this reference "Sinking *Trichodesmium* fixes nitrogen in the dark ocean" by Mar Benavdies.

Response:

We thank the Reviewer's for the reminder. We have revised the reference.

Please see Line 292-293: "*Trichodesmium* with gas vacuoles can migrate vertically to the deeper waters to optimize the uptake, utilization, and even storage of P, fulfilling the growth requirement (Sohm et al., 2011; Benavides et al., 2022).".

17. Line 343 - Regarding salinity affecting diazotrophs. Look into "Salinity as a key control on the diazotrophic community composition in the southern Baltic Sea" by Christian Reeder and "Diversity, structure, and distribution of bacterioplankton and diazotroph communities in the Bay of Bengal during the winter monsoon" by Chao Wu.

Response:

We are thankful for the Reviewer's comments. We have added further discussion about the effect of salinity on diazotrophs as below:

Line 377-379: "Salinity is another important environmental factor affecting the osmoregulation, metabolism, and community composition of diazotrophs (Zehr and Turner, 2001; Reeder et al., 2022). *Trichodesmium* was reported to be abundant in the area with high salinity (e.g., the Luzon Strait) (Wu et al, 2018), which is consistent with our analysis."

18. Figure 2 - "b" is needed. Which figures belong to which year? and secondly, legend title is needed. E.g. depths for b,c and temperature for d,e. Also legend title and unit for a

Response:

We thank the Reviewer for the comments. We have added "b", legend title and unit in Figure 2. In addition, we have revised the figure caption to clarify the sampling year for each panel in Figure 2 as below:

Line 713-717: "Figure 2. Potential temperatures, salinities and density anomalies modeled for the cruises in 2017 and 2018. (a) Plot of $\theta$-$S$ showing potential temperatures ($\theta$, °C) and salinities ($S$) of water parcels resulting from mixing of the Kuroshio and SCS water masses. Potential density anomalies ($\sigma_\theta$, kg m$^{-3}$), shown in grey lines, are imposed on the $\theta$-$S$ plot. (b–e) Profiles of depths (b, c) and temperatures (d, e) of isopycnal surface along the $\sigma_\theta$ of 23 kg m$^{-3}$ of water masses depicted for the cruises in 2017 (b, d) and 2018 (c, e)."

19. **References**

Benavides, M., Bonnet, S., Le Moigne, F.A.C. et al. Sinking Trichodesmium fixes nitrogen in the dark ocean. ISME J 16, 2398–2405 (2022). https://doi.org/10.1038/s41396-022-01289-6

Wu C, Narale DD, Cui Z, Wang X, Liu H, Xu W, Zhang G, Sun J. Diversity, structure, and distribution of bacterioplankton and diazotroph communities in the Bay of Bengal during the winter monsoon. Front Microbiol. 2022 Nov 30;13:987462. doi: 10.3389/fmicb.2022.987462. PMID: 36532434; PMCID: PMC9748438.

Reeder, C. F., Stoltenberg, I., Javidpour, J., and Löscher, C. R.: Salinity as a key control on the diazotrophic community composition in the southern Baltic Sea, Ocean Sci., 18, 401–417, https://doi.org/10.5194/os-18-401-2022, 2022.

Kendra A Turk-Kubo, Mary R Gradoville, Shunyan Cheung, Francisco M Cornejo-Castillo, Katie J Harding, Michael Morando, Matthew Mills, Jonathan P Zehr, Non-cyanobacterial

diazotrophs: global diversity, distribution, ecophysiology, and activity in marine waters, FEMS Microbiology Reviews, 2022; fuac046, https://doi.org/10.1093/femsre/fuac046

**Citation**: https://doi.org/10.5194/bg-2023-126-RC2

Response:

    We thank the Reviewer for providing the references. They are helpful and we have cited these references in the revised manuscript.

---

## Author Comment (AC2)

Manuscript bg-2023-126

"Changes in diazotrophic community structure associated with Kuroshio succession in the northern South China Sea" by Han Zhang et al.

We have taken all the comments of the Reviewers into account in the revision. Our point-by-point responses are provided below in blue fonts. Please note that all the line numbers mentioned in the response refer to the Marked-up Manuscript.

**Reviewer comments:**

**Reviewer #1 (Comments for the Author):**

1. $N_2$ fixation plays important role in food-web process, carbon sequestration and export of organic carbon to the deep ocean. Kuroshio intrusion and associated environmental alteration may profoundly affect biogeography and $N_2$ fixation rate of diazotrophs. This study demonstrated changes in diazotrophic community structure and $N_2$ fixation because of Kuroshio intrusion in the northern South China Sea (nSCS) based on two cruises in 2017 and 2018. The authors found that Trichodesmium was more abundant and $N_2$ fixation rate was higher at stations strongly affected by Kuroshio intrusion, whereas UCYN-B were more abundant at stations least affected by Kuroshio intrusion. UCYN-C and γ-proteobacteria were mainly distributed at stations moderately affected by the Kuroshio. These results suggested that diazotrophic community composition and nitrogen fixation rate in nSCS are highly regulated by Kuroshio intrusion, which will contribute to our understanding of how Kuroshio affect diazotrophic diversity and nitrogen fixation. Overall, I appreciate their excellent work. However, the presentation of manuscript (particularly Results and Discussion) needs to be improved. I will recommend consideration of its acceptance for potential publication after a minor revision.

Response:

We thank the Reviewer #1 for a very elaborated and professional review. We are thankful for the helpful comments that helped us update and improve the manuscript. We believe that the revised version is in a better form, as we did our best to address all the comments. Our point-by-point response is provided below.

2. Introduction should review what is already known about effects of Kuroshio intrusion on diazotrophs (particularly *Trichodesmium*) and nitrogen fixation in the in marginal seas of NW Pacific. Also, there is a lack of hypothese on the effects of Kuroshio intrusion on the diazotrophic community composition and $N_2$ fixation in the nSCS as well as the intrusion intensity.

Response:

We are grateful to the Reviewer for providing constructive comments. We have incorporated the Reviewer's suggestions and have made specific modifications to the Introduction accordingly. These revisions encompassed two key aspects.

First, we focused on the impacts of KI on diazotrophs (particularly *Trichodesmium*) and nitrogen fixation in the in marginal seas of western North Pacific.

Second, we focused on the hypothesis regarding the impact of Kuroshio intrusion on the composition of diazotrophic communities, nitrogen fixation, and intrusion intensity in the nSCS.

We have revised the Introduction as below:

Line 75–77: "In contrast to the well studied diazotrophic biogeography in these regions, less effort has focused on the impacts of KI on marginal seas of the northwestern Pacific (Jiang et al., 2023)."

3. Prefilteration using a 100 μm pore-size nylon mesh can remove large zooplankton, but probably remove *Trichodesmium*, particularly large-sized colonial trichomes, resulting in a potential underestimation of *Trichodesmium* abundance.

Response:

We completely agree with the reviewer that prefiltration using a 100 μm pore-size nylon mesh might potentially remove colonial trichomes of *Trichodesmium* larger than 100 μm. To be more rigorous, we clarified the rationale of prefiltration and revised the Materials and Methods and Discussion section as below:

Line 99–102: "At each station, 1.5–3 L of seawater was prefiltered through a 100-μm pore-size nylon mesh to remove large zooplankton and fish, and then filtered through a 0.22-μm pore-size 47-mm diameter polycarbonate membrane (Millipore, USA) with low pressure (<100 mm Hg pressure) for subsequent DNA extraction."

Line 293–295: "It is worth noting that, since we did not observe colonial *Trichodesmium* in our sampling stations, prefilteration using a 100-μm pore-size nylon mesh was not likely to underestimate *Trichodesmium* abundance."

4. The authors don't describe how to collect qPCR samples for daytime and nighttime in Materials and Methods, is it to take parallel samples at the same time, one of which is filtered during the daytime (nighttime) to obtain the samples, and the other is put into the nighttime (daytime) to be filtered again? Discussion also does not address in detail why the *nifH* gene abundance of the diazotrophic groups differed largely between daytime and nighttime. In general, the *nifH* gene expression in diazotrophic groups should be determined using RT-qPCR rather than qPCR.

Response:

We are thankful to the Reviewer for the comments. We have addressed your concerns regarding the collection of samples and the analysis conducted in our study.

We have revised the description in Materials and Methods as below:

Line 97–99: "Samples were collected using a Seabird SBE 911Plus rosette sampling system (Sea-Bird Electronics, USA) at 14 stations (Fig. 1 and Table S1), seven of which were chosen for both daytime and nighttime sampling, whereas the other seven stations were used only for either daytime or nighttime sampling.".

The difference in the *nifH* gene abundance of the diazotrophic groups between daytime and nighttime can be attributed to temporal and spatial variations in environmental factors across different sampling times and stations. We agree with the Reviewer that the *nifH* gene expression in

diazotrophic groups should be determined using RT-qPCR rather than qPCR. We noted that the *nifH* transcripts could be influenced by various external factors (e.g., light intensity, temperature and nutrients). In addition, the *nifH* gene expression can exhibit large variability over time within each diazotroph (Church et al., 2005; Wilson et al., 2017). Therefore, conducting comparisons across different organisms based on the *nifH* gene expression data would not be appropriate. Furthermore, given that the timing of sampling differed across stations, comparisons among these stations using the *nifH* gene expression data would not be valid, even within a single organism. We appreciate your understanding of these limitations and their implications for our study.

5. Nitrogen fixation rate and primary production are missing in Results. The authors can compare nitrogen fixation rates and primary production in the Kuroshio water, mixed water and SCS water. This analysis should be useful in further exploring the influences of changes in diazotrophic composition and nitrogen fixation induced by the Kuroshio intrusion on the carbon and nitrogen biogeochemical cycling of the nSCS as well as the implications.

Response:

We thank the Reviewer for the comments. We have added the description about nitrogen fixation and primary production rates in Results as below:

Line 228-233: "The water column-integrated rates of $N_2$ fixation ($I_{NFR}$) in the KC (129.42 ± 100.86 μmol N $m^{-2}$ $d^{-1}$) was higher than that in the nSCS (97.85 ± 6.98 μmol N $m^{-2}$ $d^{-1}$). Surface $N_2$ fixation rates (1.88 ± 0.85 nmol N $L^{-1}$ $d^{-1}$) in the nSCS was higher than those in the KC (0.60 ± 0.25 nmol N $m^{-2}$ $d^{-1}$, stn 1 to stn 4) as observed along the large-scale transect. However, the surface primary productivity was found to be similar between the nSCS (0.37 ± 0.12 μmol C $L^{-1}$ $d^{-1}$) and the KC (0.21 ± 0.11 μmol C $L^{-1}$ $d^{-1}$)."

We also have incorporated this into the Discussion in the marked-up Manuscript as below:

Line 316-321: "the nitrigen fixation rate at surface ($S_{NFR}$) in the KC was lower than that at nSCS, which may be due to the deeper nitracline in the KC inhibiting the upward transport of nutrients (Chen et al., 2008). The relatively higher nitrogen fixation rate of the entire water column ($I_{NFR}$) was observed. This special distribution pattern of $I_{NFR}$ and $S_{NFR}$ might be explained by the distinct niches of different diazotrophs, where KI-transported *Trichodesmium* typically dominates shallow warm water (< 50 m) (Jiang et al., 2015), whereas UCYN-B prefers to live in lower temperature and deeper water (50–75 m) (Moisander et al. 2010)."

6. Conclusions are not simple repetitions of the results, please revise them.

Response:

We are grateful for the Reviewer's suggestion. We have revised Conclusions as below:

Line 407-417: "In this study, we surveyed the diversity and abundance of diazotrophic phylotypes in the nSCS in two consecutive years based on molecular approaches targeting the nitrogenase gene *nifH* and associated the patterns in changes of diazotrophic community structure in the upper 100 m with the degrees of KI quantified using the isopycnal mixing model. Our study reveals a lineage-specific niche adaptation to environmental changes associated with KI in the

overall diazotrophic community structure in the nSCS. Four diazotrophic groups, *Trichodesmium*, UCYN-B, UCYN-C, and γ-proteobacteria, appear to be highly correlated with KI leading to variations in a range of physicochemical parameters. KI has a dominant dilution effect on the nutrient inventory and is responsible for the enhanced $N_2$ fixation on one hand, it also causes redistribution of the diazotrophic taxa and reallocation of the nutrients on the other hand. Neutral model simulation suggests that the diazotrophic community is more likely affected by KI primarily as a stochastic process. As KIs are projected to intensify in a future warming ocean, Kuroshio may transport diazotrophs (especially the UCYN-A) in the upstream warmer regions including SCS northward to higher latitudes, resulting in a wider distribution of $N_2$ fixation in the global ocean."

**Specific comments below:**

7. I suggest delete some unnecessary connecting adverbs in the text, such as "Moreover" and "Collectively" in Abstract.

Response:

We thank the Reviewer for the suggestion. We have removed several unnecessary connecting adverbs from the text. We have revised Abstract as below:

Line 10-23: "Kuroshio intrusion (KI) is a key process that transports water from the Western Pacific Ocean to the northern South China Sea (nSCS), where KI-induced surface water mixing often causes variations in microbial assemblages. Yet, how interannual KIs affect biogeography of diazotrophs and associated environmental factors, remains poorly characterized. Here, by quantifying the degree of KIs in two consecutive years, coupled with monitoring the diversity and distribution of nitrogenase-encoding *nifH* phylotypes with quantitative PCR and high-throughput sequencing, we show that changes in the diazotrophic community structure in the nSCS are highly correlated with KI leading to variations in a range of physicochemical parameters. Specifically, the filamentous cyanobacterium *Trichodesmium* was more abundant at stations strongly affected by KI, and hereby with deeper mixed layer, higher surface salinity, and temperature; the unicellular $N_2$-fixing cyanobacteria in group B (UCYN-B) were more abundant at stations least affected by KI and correlated with nutrient availability, whereas UCYN-C and the γ-proteobacteria were prevalent at stations moderately affected by KI. Neutral community model further demonstrated that dominant diazotrophic subcommunities were significantly affected by environmental factors in 2017 when KI was stronger compared to 2018 when KI retreated. Our analyses provide insightful evidence in the role of KI succession in shaping diazotrophic community structure primarily as a stochastic process, implying a potential region-scale redistribution of diazotrophs and nitrogen budget, given that KIs are projected to intensify in a future warming ocean."

8. L45: Please replace "UCYN-B is mostly free-living with Crocosphaera watsonii being a cultivated representative" with "UCYN-B (Crocosphaera watsonii) is mostly free-living, being a cultivated representative".

Response:

We thank the Reviewer for the suggestion. We have revised Introduction as below:

Line 43-46: "Phylogenetically distinct clades of UCYN have been detected. The "*Candidatus Atelocyanobacterium thalassa*" in group A (UCYN-A) appears to be an obligate endosymbiont of the single-celled prymnesiophyte algae (Thompson et al., 2012), whereas UCYN-B is mostly free-living and a culturable representative, *Crocosphaera watsonii,* has been successfully isolated."

9. L124-125: Light gradient should be provided.

Response:

We appreciate the Reviewers' comments and suggestions regarding our manuscript. We have added the description about light gradient as below:

Line 131-133: "Light conditions were recorded at 5 fixed depths (5, 25, 50, 60-75, 115-150 m) within the water column, corresponding to approximately 50%, 30%, 10%, 1%, and 0.1% of the surface irradiance, respectively."

10. L207: Please delete "Du et al., 2013".

Response:

We thank the Reviewer for the suggestion. We have deleted "Du et al., 2013" from the text. Please see Line 219.

11. L295: This paragraph is not directly related to nitrogen fixation. I suggest the authors emphasize the changes in physical parameters and nutrients in nSCS caused by Kuroshio intrusion, in addition the description of dynamic process. How and why these changes affect $N_2$ fixation?

Response:

We are thankful for the Reviewer's suggestion. We have integrated into the discussion about the changes in physical parameters and nutrients in nSCS caused by Kuroshio intrusion and their effects on $N_2$ fixation.

Line 354–357: "The dilution effect of KI on surface nutrient budget in nSCS could provide favorable ecological niches for diazotrophs, limiting other phytoplankton which can not conduct $N_2$ fixation (Zehr & Capone 2020). This might partially elucidate why KI has the potential to enhance N2 fixation rates in the nSCS.".

12. L314-315: In addition to the results of Kao et al., 2012, how diazotrophs contributed nitrogen budget and primary/new production.

Response:

We thank the Reviewer for the comment. According to previous studies conducted by Chen et al (2014) and Wu et al (2018), it has been suggested that unicellular diazotrophs play a significant role in $N_2$ fixation in the South China Sea (SCS) and the Kuroshio. Wu et al (2018) reported that unicellular diazotrophs contribute approximately 75% of $N_2$ fixation in the northern SCS and the Kuroshio. Similarly, Chen et al (2014) demonstrated that unicellular diazotrophs contribute 65% and 50% of $N_2$ fixation in the SCS and the Kuroshio, respectively.The contribution of $N_2$ fixation to primary production has been found to be less than 20% in studies conducted by Lu et al (2019) and Wu et al (2018). The estimated amount of new nitrogen introduced by *Trichodesmium*

contributed up to 0.14% of the total primary production and 0.41% of the new production in the Luzon Strait (Wu et al 2018). We revised the text as below:

Line 343–345: "N$_2$ fixation in the nSCS was estimated to be $\sim$20 mmol N m$^{-2}$ yr$^{-1}$, which accounts for less than 10% of the current nutrient inventory variation ($\sim$250 mmol N m$^{-2}$ yr$^{-1}$) (Kao et al 2012). Unicellular diazotrophs were reported to contribute more than 50% of N$_2$ fixation (Chen et al 2014, Wu et al 2018). "

13. L318-319: These values of 75.98 ±48.77 μmol N m$^{-2}$ d$^{-1}$ and 74.98 ± 26.55 μmol N m$^{-2}$ d$^{-1}$ were the average NFR in nSCS during 2017 and 2018, respectively.

Response:

We thank the Reviewer for the comment. We have revised the text as below:

Line 351-352: "The KI was conspicuously responsible for the higher water column-integrated rates of N$_2$ fixation (I$_{NFR}$) in KC than in nSCS".

14. Table S4: I suggest remove Table S4 from supplementary materials to the text.

Response:

We thank the Reviewer for the suggestion. We have removed Table S4 from supplementary materials to the text (Table 1).

15. Table S5: Why are there two sampling times for the same station in Table S5, is it to compare the effects of daytime and nighttime on diazotrophs? If so, RT-qPCR should have been used to determine *nifH* gene expression in different diazotrophic groups.

Response:

We thank the Reviewer for the question. In Table S4 (Table 5 in the previous version), we removed the *nifH* gene copies data of nighttime. We primarily focused on surface daytime data to conduct our subsequent analysis (including determination of abundant and rare taxa, correlation of microbial communities with environmental factors and geographical distance, and diazotrophic community assembly modeling) in the main text. We agree with the Reviewer that anlysis on the effects of daytime and nighttime on diazotrophs should be based on the *nifH* gene expression data using RT-qPCR. However, this might not be the main purpose of this study.

Furthermore, we conducted quantitative and high-throughput sequencing experiments at the DNA level due to the potential influence of external factors on gene expression at the transcriptional level. In order to maintain consistency and minimize such influences, we specifically selected samples collected during the day for our analysis. We appreciate your feedback and hope that this explanation addresses your concerns satisfactorily.

16. Figure 1: Please give the full names of nSCS an KI.

Response:

We thank the Reviewer for the suggestion. We have give the full names of nSCS and KI in the caption of Figure 1.

17. Figure 2: "(b)" is lacking.

Response:

We appreciate the Reviewer's comment. We have added "(b)" in Figure 2.

18. Figure 6: Please give the full name of RDA. In addition, the relative contribution of different parameters to variation in diazotrophic community composition needs to be quantified.

Response:

We thank the Reviewer for the suggestion. We have given the full name of Redundancy analysis in the caption of Figure 6. In addition, the relative contribution of different parameters to variation in diazotrophic community composition have been quantified in supplementary materials Table S8, S9.

19. The station numbers in Fig. 3 and Table S5 did not match. Figure 3 and Table S5 showed that the abundance of Richelia in Kuroshio water was higher than that in SCS water, which is similar to the findings of Tuo et al (2014). Chen et al. (2014) has showed that nitrogen fixation rate was much higher in Kusoshio than in nSCS during warm seasons. Also, they found that relative contribution (59%) of filamentous diazotrophs (>10 or 20 mm, mostly *Trichodesmium* and *Richelia*) to nitrogen fixation was higher than that (41%) of unicellular filamentous during warm seasons. Similarly, result of Jiang et al. (2023) suggested that Kuroshio intrusion stimulated growth of *Trichodesmium* and enhanced nitrogen fixation in the East China Sea during summer. Therefore, the present study on diazotrophic abundance and nitrogen fixation was highly consistent with earlier studies in marginal seas of NW Pacific suffered from intrusion of Kuroshio. I suggest the authors analyze more deeply about the mechanism of Kuroshio intrusion on the community composition of diazotrophs and nitrogen and carbon fixation.

Response:

We appreciate the Reviewer's valuable comments. Figure 3 presented the surface daytime abundance of the ten major diazotrophic groups, as determined by the TaqMan qPCR assay of the *nifH* gene copies. In Table S4 (Table 5 in the previous version), we had included the diel patterns in abundance of the targeted diazotrophic groups. We apologized for any confusion caused by the absence of nighttime data in Table S4. We have revised the Table S4 in supplementary materials. We have revised the text as below:

Line 286-288: "The high abundance of *Trichodesmium* was previously reported in the East China Sea affected by KI during summer (Jiang et al., 2023). This may be attributed to *Trichodesmium*'s higher temperature preference and the coastal input of iron when Kuroshio flows past some islands (Cheung et al., 2017)."

Table S8. The importance components, eigenvalue proportion explained and cumulative proportion in redundancy analysis (RDA)

| Components | Eigenvalue | Proportion | Cumulative |
|---|---|---|---|
| RDA1 | 0.1640 | 0.5770 | 0.5770 |
| RDA2 | 0.0615 | 0.2164 | 0.7935 |
| RDA3 | 0.0309 | 0.1086 | 0.9021 |
| RDA4 | 0.0121 | 0.0426 | 0.9447 |
| RDA5 | 0.0065 | 0.0228 | 0.9675 |
| RDA6 | 0.0025 | 0.0087 | 0.9761 |
| RDA7 | 0.0011 | 0.0040 | 0.9802 |
| RDA8 | 0.0009 | 0.0031 | 0.9833 |
| RDA9 | 0.0003 | 0.0012 | 0.9845 |
| RDA10 | 0.0001 | 0.0003 | 0.9847 |
| RDA11 | 0.0000 | 0.0001 | 0.9848 |
| RDA12 | 0.0000 | 0.0000 | 0.9849 |
| RDA13 | 0.0043 | 0.0151 | 1 |

Table S9. The contribution of variables in the environments factors matrix and major diazotrophic groups to the variation in the species matrix explained by the RDA. The environmental factors used in this study are shown in Table S4.

| | | RDA1 | RDA2 |
|---|---|---|---|
| Environments factors | SST | -0.0952 | 0.5214 |
| | SSS | -0.1040 | 0.7065 |
| | MLD | 0.1829 | 0.4228 |
| | DCM | 0.0851 | 0.8747 |
| | $I_{DIN}$ | -0.8404 | -0.2569 |
| | $I_{DIP}$ | -0.9077 | -0.1455 |
| | Nit | -0.8394 | 0.1058 |
| | $I_{NFR}$ | -0.3828 | 0.4466 |
| | $I_{PP}$ | -0.3233 | -0.5956 |
| | $S_{NFR}$ | -0.6608 | 0.0791 |
| | $S_{PP}$ | -0.4627 | -0.5505 |
| | $R_{K\_100}$ | 0.1800 | 0.6767 |
| Major diazotrophic groups | *Trichodesmium* | 0.0110 | 0.5879 |
| | UCYN-B | -0.7612 | -0.1394 |
| | UCYN-C | 0.5982 | -0.1114 |
| | γ-proteobacteria | 0.2126 | -0.1731 |

---

## Author Response (AR1)

Manuscript bg-2023-126

"Changes in diazotrophic community structure associated with Kuroshio succession in the northern South China Sea" by Han Zhang et al.

We have taken all the comments of the Reviewers into account in the revision. Our point-by-point responses are provided below in blue fonts. Please note that all the line numbers mentioned in the response refer to those in the Marked-up Manuscript.

General comments:

Reviewer (Comments for the Author):

Zhang et al., 2023 show how the diazotrophic community in the northern South China Sea (nSCS) responds to intrusion by Kuroshio current waters by measuring nifH abundances via qPCR and high throughput sequencing, nitrogen fixation rates, and the degree of Kuroshio intrusion (KI) over two cruises in 2017 and 2018. The big picture question the authors address is an important one, namely how do physical changes in a marine ecosystem affect microbial diversity? The authors show that *Trichodesmium* is more abundant at stations with more KI and that UCYN-B is more abundant at stations with low KI. In addition, the authors show that UCYN-C and gamma-proteobacteria are more prevalent at stations moderately affected by KI. The authors also perform statistical tests to assess the degree environmental factors affect the diazotrophic community structure. Their tests suggests that environmental factors have a bigger affect on diazotrophic community structure in 2017, when KI is strong, than in 2018 when KI is weak.

I have several minor comments but no major revisions and recommend the paper to be accepted with minor revisions. All line numbers refer to the preprint visible on EGUsphere.

Response:

We thank the Reviewer for the comments and consideration of this manuscript for acceptance. We hope that our point-by-point responses below have addressed his/her concern.

**General comment**

I found it difficult to understand what the Mantel tests vs. the NCM were testing. The abstract and Figure 8 say that the neutral community model is being used to test the effect of environmental variables on diazotrophic community composition. However, in the methods and elsewhere, the Mantel tests are described as the way that relationship is assessed. I think the easiest way to solve this would be to add an explanation in section 2.9 that describes how the NCM was used to assess the variation due to environmental and spatial factors.

Response:

Mantel tests and the Neutral Community Model (NCM) serve distinct purposes and are applied to investigate different aspects of microbial communities. Mantel tests are often used to examine the correlation between distance matrices that reflect the differences in microbial community composition between different samples, such as the Bray-Curtis dissimilarity or UniFrac distances. The Mantel test can also be used to assess how microbial community the correlation composition relates to specific environmental factors or spatial patterns within a given ecosystem. The NCM, on

the other hand, is a theoretical framework used to explore the role of ecological drift and dispersal limitation in shaping microbial community structure. This model assumes that microbial community assembly is primarily governed by random processes, such as birth, death, and migration, rather than deterministic ecological interactions. Together, these approaches contribute to our understanding of the factors driving the structure and dynamics of microbial communities in various environments.

We are grateful for the Reviewer's comment. We agree with the Reviewer that add an explanation in section 2.9 that describes how the NCM was used to assess the variation due to environmental and spatial factors. Therefore, we have added further description about NCM in section 2.9 as below:

Line 213-217: "To assess the role of KI in shaping diazotrophic community assembly, we fit the occurrence frequency of diazotrophic taxa across the wider metacommunity to the neutral community model (NCM), a theoretical framework used to explore the potential impact of stochastic processes (Sloan et al., 2007). Calculation of 95% confidence interval was done using 1,000 bootstrap replicates, and the overall fit to the NCM was indicated by the parameter $R^2$. All data analyses were processed in R (version 4.1.3) (http://www.r-project.org).".

**Line by line comments**

Title - I agree with the other reviewers that it should be Changes not Change

Response:

We thank the Reviewer for the suggestion, and we have revised the Title as "Changes in diazotrophic community structure associated with Kuroshio succession in the northern South China Sea".

Line 35 - This phrasing is awkward. I would rephrase "The current state of marine $N_2$ fixation study ..." to be "The field of marine $N_2$ fixation is changing dramatically."

Response:

Thanks for the suggestion and we have rephrased it. Please refer to Line 35.

Line 35 : "The field of marine $N_2$ fixation is changing dramatically.".

Lines 39 - 40 - I would also change the phrasing here. I would replace "Such previously established concept" with "This previously established concept ... "

Response:

Thanks for the suggestion and we have replacedd "such" with "this"..

Line 39-40 : "This previously established concept was challenged by the discovery of unicellular…".

Lines 35-52 - Since UCYN-C is discussed in the abstract, I would explicitly mention it when the new diatom-diazotroph symbioses discovered by Schvarcz, 2022 are discussed. I would also mention another way that the field of marine $N_2$ fixation is changing, which is the discovery of widespread, high coastal or continental shelf $N_2$ fixation (Tang et al., 2019).

Response:

We thank the Reviewer for the comment. We have extended the description about new diatom-diazotroph symbioses and the biogeography of marine $N_2$ fixation as below:

Line 49-56: "Particularly, diazotrophs genomes reconstructed from global metagenomic data have extended the PCR-based amplicon surveys revealing new diatom-diazotroph symbioses (Schvarcz et al., 2022) and new species of non-cyanobacterial diazotrophs (NCDs) (Bombar et al., 2016; Delmont et al., 2018).The Rhopalodiaceae diatoms have been repeatedly isolated from the subtropical North Pacific Ocean, with the endosymbionts having *nifH* gene sequences similar to those of free-living UCYN-C cyanobacteria (Schvarcz et al., 2022). NCDs have been reported to be ubiquitous in ocean ecosystems and contribute to global marine $N_2$ fixation (Moisander et al., 2014; Chakraborty et al., 2021; Turk-Kubo et al., 2022). Recently, widespread and high $N_2$ fixation was also discovered in the coastal or continental shelf (Tang et al., 2019).".

Line 94 - I agree with the previous reviewer comment that a 100 micron pre-filter could reduce *Trichodesmium* abundances. I would add in a comment mentioning this if you have not already.

Response:

We completely agree with the reviewer that prefiltration using a 100 μm pore-size nylon mesh might potentially remove colonial trichomes of *Trichodesmium* larger than 100 μm. To be more rigorous, we clarified the rationale of prefiltration and revised Materials and Methods and Discussion sections as below:

Line 100-102: "At each station, 1.5-3 L of seawater was prefiltered through a 100-μm pore-size nylon mesh to remove large zooplankton and fish, and then filtered through a 0.22-μm pore-size 47-mm diameter polycarbonate membrane (Millipore, USA) with low pressure (<100 mm Hg pressure) for subsequent DNA extraction.".

Line 291-293: "It is worth noting that, since we did not observe colonial *Trichodesmium* in our sampling stations, prefilteration using a 100 μm pore-size nylon mesh was not likely to underestimate *Trichodesmium* abundance.".

Line 173 - Please include a reference for your merging criteria.

Response:

We thank the Reviewer for the comment. We have added the reference about the merging criteria as below:

Line 183: "…ii) Sequences having ≥ 10 bp overlap but ≤ 2 bp mismatch were merged (Zhang et al., 2023)…".

Line 177 - Please also include a reference for the RDP classifier algorithm.

Response:

We thank the Reviewer for the comment. We determined relative abundance of OTU using UPARSE with a 97% similarity cutoff according to Quince et al. (2011).

Line 184-189: "Operational taxonomic units (OTUs) were clustered with a 97% similarity cutoff using UPARSE (version 7.1) (http://drive5.com/uparse/) with a novel 'greedy' algorithm that performs chimera filtering and OTU clustering simultaneously (Quince et al., 2011). OTUs were annotated down to genus level using a formatted *nifH* gene database (https://github.com/moyn413/nifHdada2) that was updated in 2023 according to NCBI and Zehr Lab (https://www.jzehrlab.com/*nifH*).".

Line 190 - It is not clear how you define spatial vs. environmental factors. Please state all spatial and environmental factors you test explicitly.

Response:

We thank the Reviewer for the comment. We have added the description about the spatial vs. environmental factors as below:

Line 202-204: "The environmental factors are summarized in Table 1. All environmental parameters, except pH, were log (X+1)-transformed to improve homoscedasticity and normality for multivariate statistical analyses and calculation of the Euclidean distances between samples.".

Line 204-205: A set of spatial variables based on the longitude and latitude coordinates of each sampling station were calculated following the approach of the principal coordinates of neighbor matrices (PCNMs) analysis (Borcard and Legendre, 2002).".

Line 196 - Please also include a citation to justify the choice of a VIF threshold of < 20.

Response:

We thank the Reviewer for the comment. We have added the citation for the choice of a VIF threshold of < 20 as below:

Line 206-207: "To avoid collinearity among factors, explanatory environmental factors with the highest variance inflation factor (VIF) were eliminated until all VIF values were lower than 20 (Blanchet et al., 2008; Chen et al., 2019).".

Line 202 - I do not understand the two phrases used to describe what the NCM is assessing the relationship between "the occurrence frequency of diazotrophic taxa" and "their relative abundance across the wider metacommunity". I would rephrase to more clearly explain the difference between "the occurrence frequency of diazotrophic taxa" and "their relative abundance across the wider metacommunity".

Response:

We thank the Reviewer for the comment. We have rephrased to more clearly explain the difference between "the occurrence frequency of diazotrophic taxa" and "their relative abundance across the wider metacommunity". The frequency of occurrence of diazotrophic taxa refers to how often these organisms are present in a given context, while the relative abundance across the wider metacommunity indicates the proportion or percentage of these taxa in comparison to the entire community. In simpler terms, the occurrence frequency focuses on how often they appear, while relative abundance considers their prevalence within the broader community. we added the description about the NCM in Methods 2.9 as below:

Line 213-217: "To assess the role of KI in shaping diazotrophic community assembly, the occurrence frequency of diazotrophic taxa across the wider metacommunity was fitted to the neutral community model (NCM), a theoretical framework used to explore the potential impact of stochastic processes (Sloan et al., 2007). Calculation of 95% confidence interval was done using 1,000 bootstrap replicates, and the overall fit to the NCM wasindicated by the parameter $R^2$. All data analyses were processed in R (version 4.1.3) (http://www.r-project.org).".

Lines 218 - 219 - This sentence should be in the methods not in the results.
Response:

We thank the Reviewer for the comment. We have removed the sentence to Methods as below:
Line 163-165: "When the standard clone was diluted 10 times, the corresponding Ct value increased by about 3.3–3.4 units, indicating that the PCR amplification efficiencies among all replicates were between 90% and 100% ($R^2 > 0.99$, Fig. S1).".

Lines 226 - 230 - I didn't see any tables where the relative abundances were clearly stated. I would make a new SI table with the same format as Table S5 that has the relative abundances for the 10 species investigated via qPCR. It is also hard to evaluate the authors' claims about day vs. nighttime abundances since many of the times (19:00, 19:30, 20:00 are on the border of day and night). I would explicitly state with a D or N in Table S5 and the new table which times are day and night. Please also say if times are local time or another timezone.
Response:

We thank the Reviewer for the comment. We apologize for the confusion caused. We have made a new SI table (Table S5) with the same format as Table S4 (Table S5 in the previous version) that has the relative abundances for the 10 species investigated via qPCR. In Table S4 (Table S5 in the previous version) and Table S5, we removed the *nifH* gene copies data of nighttime and focused on surface daytime data in the main text. we would like to clarify that the times mentioned in our study are presented in the local time zone.

Line 232 - This statement is misleading because there is quite a bit of variability across stations, for example stn 1 vs. 12 vs. 4 in 2017 (Table S6) . I would instead describe that there is considerable

station to station variability but that across all stations cyanobacteria and gamma proteobacteria are at approximately equal abundances.

Response:

We thank the Reviewer for the comment. We have revised the statement as below:

Line 246-249: "For DNA-based evaluation of *nifH* amplicons with NGS, the majority (>85%) of the *nifH* amplicons belonged to cyanobacteria and γ-proteobacteria (Table S6). There was considerable station to station variability but across all stations cyanobacteria and γ-proteobacteria were at approximately equal abundances in 2017. In contrast, a higher proportion of cyanobacteria (64%) was detected in 2018 compared to γ-proteobacteria (23%).".

Line 233-Here, I think there also needs to be an acknowledgement that at some stations *Trichodesmium* is much different than 58% of the cyanobacterial *nifH* abundances. For example at stn 1 in 2017 - *Trichodesmium* is ~75%. Like above, I would say that there is considerable variability but that overall *Trichodesmium* is 58% of cyanobacterial abundances.

Overall comment here - describing the overall pattern while acknowledging station to station variability will allow you to transition to the next section - because the station to station variability is consistent with what your correlation analyses show. For example, Stn 1 which is mostly Kuroshio waters has predominantly *Trichodesmium*.

Response:

We thank the Reviewer for the comment. We have revised the statement as below:

Line 249-251: "Overall, *Trichodesmium* and UCYN were approximately 58% and 42%, respectively, of cyanobacterial abundances despite station to station variability. The UCYN-C (22%) and UCYN-B (18%) were the domiant UCYN sublineages in 2017 and 2018, respectively (Table S6)."

Line 242 and/or Figure 5 - In Figure 5, $I_{NFR}$ not $S_{NFR}$ is in the same cluster as SST, SSS, DCM etc. Either there is a typo at line 242 or a typo in Figure 5.

Response:

We thank the Reviewer for the comment. We have changed "$S_{NFR}$" to "$I_{NFR}$" in Line 258.

Line 246 and/or Figure 5 - In Figure 5, $S_{PP}$ looks like it is in a separate cluster from $I_{DIN}$, $I_{DIP}$, and Nit. If you meant the previous fork in the tree, I would list all six variables that are not in the strong KI cluster.

Response:

We thank the Reviewer for the comment. We have revised the description as below:

Line 259-261: "…, whereas the depth-integrated dissolved inorganic phosphorus ($I_{DIP}$), dissolved inorganic nitrogen ($I_{DIN}$), primary production ($I_{PP}$), surface nitrogen fixation rate ($S_{NFR}$), surface primary production ($S_{PP}$) and the depth of nitracline (Nit) with weak KI (Fig. 5).".

Line 257 - Based on figure 8, the variation due to only environmental factors is 0.13 in 2017. The variation due to only environmental factors as well as the variation due to both spatial and environmental factors is 0.33.

Response:

We thank the Reviewer for the comment. We have carefully addressed the comments and made the necessary revisions accordingly. The relative contribution of both components was explained by pure spatial variables, pure environmental variables, spatial variables, environmental variables and the combined effects of both space and environment. In the manuscript, our aim is to illustrate the influence of environmental or spatial factors on the variations in diazotrophic community structure. We regret any confusion arising from the statement "solely by environmental factors." To address this, we have removed the description pertaining to the variation in diazotrophic community structure explained exclusively by environmental factors, as indicated below:

Line 271-274: "The result of VPA further confirmed that the variation in diazotrophic community structure explained by environmental factors was higher in 2017 than in 2018 (0.33 vs. 0.22), whereas the influence of spatial factors was limited and similar between the two years (0.23 vs. 0.26) (Figs. 8a and 8b).".

Lines 275-276 - I would rephrase this sentence, because the 2006 paper does not show what this sentence describes. I would rephrase to "Additionally the relatively lower abundance of UCYN-B in Kuroshio waters may be because *Trichodesmium* has greater genetic resources which allow it to outcompete UCYN-B for P."

Response:

We thank the Reviewer for the comment. We have rephrased the description as below:

Line 289-291: "Additionally, the relatively lower abundance of UCYN B in Kuroshio may be due to the greater genetic resources of *Trichodesmium*, which allows it to outcompete UCYN-B for phosphorus (P) (Dyhrman et al., 2006)."

Line 285 - I think this point should be nuanced more. It looks like UCYN-C was not detected at all at station 10 and in station 9 was detected at lower levels than in 2017.

Response:

We thank the reviewer for the reminder. We have checked our data, and updated the Figure 2, Tables S4 and S5 accordingly.

Line 302-303: "Our data show that UCYN-C was among the least abundant diazotrophs in 2017 when KI was strong, but its relative abundance was 1–2 orders of magnitude higher in 2018 when KI was weak (Fig. 3).".

[Figure]

Figure 3. Surface daytime abundance of the ten major diazotrophic groups determined based on TaqMan qPCR assay of the *nifH* gene copies in samples collected in 2017 (a) and 2018 (b). Sector radius represents log-transformed absolute abundance of the *nifH* gene copies, with the angle indicating the relative proportion of each group. For the stations sampled in both years, the station number in 2018 are indicated with an asterisk sign (*).

Lines 288-293 - This paragraph is very confusing. It is difficult to figure out what argument the authors are trying to make.

Response:

We thank the Reviewer for the comment. We regret any confusion arising from the statement. Here we try to explain the differences in the relative abundances of UCYN-C and NCDs between 2017 and 2018, and attribute this primarily to the variations in the adaptive capabilities of different $N_2$-fixing phylotypes in response to KI. Additionally, we conclude that KI influences the distribution of these diazotrophs mainly as a stochastic factor.

Line 331 - I think predominated is a more accurate word than was distributed.

Response:

We thank the Reviewer for the comments. We have replaced the "distributed" to "predominated".

Line 356-357: "Specifically, *Trichodesmium* was predominated at the stations with high SSS and SST and deep MLD...".

Line 335 - This difference between selective and neutral is not discussed before. It isn't clear to me what makes spatial factors neutral. Please add a section to the methods that defines this.

Response:

We thank the Reviewer for the comment. We have added a section to the methods that defines the difference between selective and neutral processes.

Line 350-356: "Differences in the types and abundances of diazotrophs may arise through selection-driven (deterministic) and/or non-selection driven (stochastic) processes. Deterministic processes may drive differences between communities through species sorting in response to local environmental conditions, while stochastic processes may generate variation through a combination of other assembly processes including dispersal limitation, community drift and speciation (Hughes et al., 2008; Hanson et al., 2012). These stochastic processes—which we define here as 'neutral' processes—are considered in ecological neutral theories and are predicted to produce variation in community structure through space without needing to invoke the actions of selection.".

Line 338 - Another interpretation could be that the rare taxa are not constrained by environmental variables at all (i.e. rare taxa are always rare in this environment)- a really interesting result. I would further discuss this.

Response:

We thank the Reviewer for the comment. We have added the description about the importance of these new groups as below:

Line 363-366: "This result suggests that the AT* and RT subcommunities may response differently to environmental variables, or that RT may not bet constrained by environmental variables possibly due to their low growth rate, low competition potential, and narrow resource range (Pedros-Ali., 2006; Reveillaud et al., 2014).".

Line 339 - I would replace featuring with "correlated with" or "connected to".

Response:

We thank the Reviewer for the comment. We have replaced "featuring" with "correlated with".

Line 365-366: "Among the environmental parameters correlated with KI, temperature is a major factor controlling the distribution of diazotrophs.".

Line 362 - Please explain more - why especially UCYN-A?

Response:

We thank the Reviewer for the comment. To avoid the confusion, we have delete "(especially the UCYN-A)" from sentence.

Line 387-389: "Since $N_2$ fixation by marine diazotrophs has been proposed as one of the potential negative feedback mechanisms corresponding to ocean warming (Sohm et al., 2011), Kuroshio may transport diazotrophs in the upstream warmer regions including SCS northward to higher latitudes, resulting in a wider distribution of $N_2$ fixation in the global ocean.".

Lines 379-380 - There is a positive correlation between NFR and KI at just one of the strong KI stations so I don't think this claim is justified.

Response:

We thank the Reviewer for the comment. To avoid the confusion, we have delete "and is responsible for the enhanced $N_2$ fixation" from the sentence.

Line 402-404: "KI has a dominant dilution effect on the nutrient inventory on one hand, it also causes redistribution of the diazotrophic taxa and reallocation of the nutrients on the other hand."

Lines 383-384 - There is already $N_2$ fixation in temperate waters that is high - see Tang et al 2019 paper. I think the bigger picture importance of the paper is better said along the lines of the last sentence in the abstract.

Response:

We thank the Reviewer for the comment. We have revised the sentence as below:

Line 404-406: "As KIs are projected to intensify in a future warming ocean, Kuroshio may potentially cause a wider distribution of diazotrophs at high latitudes".

Figure 2 - Please make the numbers on the contours and axes bigger. They are very very small and hard to read right now. Please also clarify, is the data in panel A the real data used to model the points in panels b-e? I would state this in the legend.

Response:

We thank the Reviewer for the comment. We have made the numbers on the contours and axes bigger. The data in (a) are the real data used to model the points in panels (b-e).

[Figure]

Figure 2. Potential temperatures, salinities and density anomalies modeled for the cruises in 2017 and 2018. (a) Plot of *θ-S* showing potential temperatures (*θ*, ℃) and salinities (S, PSU) of water parcels resulting from mixing of the Kuroshio and SCS water masses. Potential density anomalies ($\sigma_\theta$, kg m$^{-3}$), shown in grey lines, are imposed on the *θ-S* plot. (b–e) Profiles of depths (b, c) and temperatures (d, e) of isopycnal surface along the $\sigma_\theta$ of 23 kg m$^{-3}$ of water masses depicted for the cruises in 2017 (b, d) and 2018 (c, e). The data points in (a) are the real data used to create the contour plots in panels b–e.

**References**

Blanchet, F. G., Legendre, P., and Borcard, D.: Forward selection of explanatory variables, Ecology, 89, 2623-2632, https://doi.org/10.1890/07-0986.1, 2008.

Chen, W., Ren, K., Isabwe, A., Chen, H., Liu, M., and Yang, J.: Stochastic processes shape microeukaryotic community assembly in a subtropical river across wet and dry seasons, Microbiome, 7, 138, https://doi.org/10.1186/s40168-019-0749-8, 2019.

Hanson, C. A., Fuhrman, J. A., Horner-Devine, M. C., and Martiny, J. B. H.: Beyond biogeographic patterns: Processes shaping the microbial landscape, Nat Rev Microbiol, 10, 497-506, https://doi.org/10.1038/nrmicro2795, 2012.

Hughes, A. R., Inouye, B. D., Johnson, M. T. J., Underwood, N., and Vellend, M.: Ecological consequences of genetic diversity, Ecol Lett, 11, 609-623, https://doi.org/10.1111/j.1461-0248.2008.01179.x, 2008.

Pedrós-Alió, C.: Marine microbial diversity: can it be determined? Trends Microbiol., 14, 257-263, https://doi.org/10.1016/j.tim.2006.04.007, 2006.

Quince, C., Lanzen, A., Davenport, R. J., and Turnbaugh, P. J.: Removing noise from pyrosequenced amplicons, BMC Bioinformatics, 12, 38, https://doi.org/10.1186/1471-2105-12-38, 2011.

Reveillaud, J., Maignien, L., Eren, A. M., Huber, J. A., Apprill, A., Sogin, M. L., and Vanreusel, A.: Host-specificity among abundant and rare taxa in the sponge microbiome, The ISME Journal, 8, 1198-1209, https://doi.org/10.1038/ismej.2013.227, 2014.

Tang, W., Wang, S., Fonseca-Batista, D., Dehairs, F., Gifford, S., Gonzalez, A. G., Gallinari, M., Planquette, H., Sarthou, G., and Cassar, N.: Revisiting the distribution of oceanic N-2 fixation and estimating diazotrophic contribution to marine production, Nat. Commun., 10, https://doi.org/10.1038/s41467-019-08640-0, 2019.

Zhang, M., Delgado-Baquerizo, M., Li, G., Isbell, F., Wang, Y., Hautier, Y., Wang, Y., Xiao, Y., Cai, J., Pan, X., and Wang, L.: Experimental impacts of grazing on grassland biodiversity and function are explained by aridity, Nat. Commun., 14, 5040, https://doi.org/10.1038/s41467-023-40809-6, 2023.

---

## Editor Decision (ED1)

**"Change in diazotrophic community structure associated with Kuroshio succession in the northern South China Sea"**

Zhang et al., 2023

**Review**

Zhang et al., 2023 show how the diazotrophic community in the northern South China Sea (nSCS) responds to intrusion by Kuroshio current waters by measuring *nifH* abundances via qPCR and high throughput sequencing, nitrogen fixation rates, and the degree of Kuroshio intrusion (KI) over two cruises in 2017 and 2018. The big picture question the authors address is an important one, namely how do physical changes in a marine ecosystem affect microbial diversity? The authors show that *Trichodesmium* is more abundant at stations with more KI and that UCYN-B is more abundant at stations with low KI. In addition, the authors show that UCYN-C and gamma-proteobacteria are more prevalent at stations moderately affected by KI. The authors also perform statistical tests to assess the degree environmental factors affect the diazotrophic community structure. Their tests suggests that environmental factors have a bigger affect on diazotrophic community structure in 2017, when KI is strong, than in 2018 when KI is weak.

I have several minor comments but no major revisions and recommend the paper to be accepted with minor revisions. All line numbers refer to the preprint visible on EGUsphere.

**General comment**

I found it difficult to understand what the Mantel tests vs. the NCM were testing. The abstract and Figure 8 say that the neutral community model is being used to test the effect of environmental variables on diazotrophic community composition.

However, in the methods and elsewhere, the Mantel tests are described as the way that relationship is assessed.

I think the easiest way to solve this would be to add an explanation in section 2.9 that describes how the NCM was used to assess the variation due to environmental and spatial factors.

**Line by line comments**

Title - I agree with the other reviewers that it should be Changes not Change

Line 35 - This phrasing is awkward. I would rephrase "The current state of marine N2 fixation study . . ." to be "The field of marine N2 fixation is changing dramatically."

Lines 39 - 40 - I would also change the phrasing here. I would replace "Such previously established concept" with "This previously established concept . . . "

Lines 35 - 52 - Since UCYN-C is discussed in the abstract, I would explicitly mention it when the new diatom-diazotroph symbioses discovered by Schvarcz, 2022 are discussed. I would also mention another way that the field of marine N2 fixation is changing, which is the discovery of widespread, high coastal or continental shelf N2 fixation (Tang et al., 2019).

Line 94 - I agree with the previous reviewer comment that a 100 micron pre-filter could reduce *Trichodesmium* abundances. I would add in a comment mentioning this if you have not already.

Line 173 - Please include a reference for your merging criteria.

Line 177 - Please also include a reference for the RDP classifier algorithm.

Line 190 - It is not clear how you define spatial vs. environmental factors. Please state all spatial and environmental factors you test explicitly.

Line 196 - Please also include a citation to justify the choice of a WIF threshold of < 20.

Line 202 - I do not understand the two phrases used to describe what the NCM is assessing the relationship between. I would rephrase to more clearly explain the difference between "the occurrence frequency of diazotrophic taxa" and "their relative abundance across the wider metacommunity"

Lines 218 - 219 - This sentence should be in the methods not in the results.

Lines 226 - 230 - I didn't see any tables where the relative abundances were clearly stated. I would make a new SI table with the same format as Table S5 that has the relative abundances for the 10 species investigated via qPCR. It is also hard to evaluate the authors' claims about day vs. nighttime abundances since many of the times (19:00, 19:30, 20:00 are on the border of day and night). I would explicitly state with a D or N in Table S5 and the new table which times are day and night. Please also say if times are local time or another timezone.

Line 232 - This statement is misleading because there is quite a bit of variability across stations, for example stn 1 vs. 12 vs. 4 in 2017 (Table S6) . I would instead describe that there is considerable station to station variability but that across all stations cyanobacteria and gamma proteobacteria are at approximately equal abundances.

Line 233 - Here, I think there also needs to be an acknowledgement that at some stations *Trichodesmium* is much different than 58% of the cyanobacterial nifH abundances. For example at stn 1 in 2017 - Tricho is ~75%. Like above, I would say that there is considerable variability but that overall Tricho is 58% of cyanobacterial abundances.

Overall comment here - describing the overall pattern while acknowledging station to station variability will allow you to transition to the next section - because the station to station variability

is consistent with what your correlation analyses show. For example, Stn 1 which is mostly Kuroshio waters has predominantly Tricho.

Line 242 and/or Figure 5 - In Figure 5, I NFR not S NFR is in the same cluster as SST, SSS, DCM etc. Either there is a typo at line 242 or a typo in Figure 5.

Line 246 and/or Figure 5 - In Figure 5, SPP looks like it is in a separate cluster from I DIN, I DIP, and Nit. If you meant the previous fork in the tree, I would list all six variables that are not in the strong KI cluster.

Line 257  - Based on figure 8, the variation due to only environmental factors is 0.13 in 2017. The variation due to only environmental factors as well as the variation due to **both** spatial and environmental factors is 0.33

Lines 275 - 276 - I would rephrase this sentence, because the 2006 paper does not show what this sentence describes. I would rephrase to "Additionally the relatively lower abundance of UCYN-B in Kuroshio waters may be because Trichodesmium has greater genetic resources which allow it to outcompete UCYN-B for P."

Line 285 - I think this point should be nuanced more. It looks like UCYN-C was not detected at all at station 10 and in station 9 was detected at lower levels than in 2017.

Lines 288 - 293 - This paragraph is very confusing. It is difficult to figure out what argument the authors are trying to make.

Line 331 - I think predominated is a more accurate word than was distributed.

Line 335 - This difference between selective and neutral is not discussed before. It isn't clear to me what makes spatial factors neutral. Please add a section to the methods that defines this.

Line 338 - Another interpretation could be that the rare taxa are not constrained by environmental variables at all (i.e. rare taxa are always rare in this environment)- a really interesting result. I would further discuss this.

Line 339 - I would replace featuring with "correlated with" or "connected to".

Line 362 - Please explain more - why especially UCYN-A?

Lines 379 - 380 - There is a positive correlation between NFR and KI at just one of the strong KI stations so I don't think this claim is justified.

Lines 383 - 384 - There is already N2 fixation in temperate waters that is high - see Tang et al 2019 paper. I think the bigger picture importance of the paper is better said along the lines of the last sentence in the abstract.

Figure 2 - Please make the numbers on the contours and axes bigger. They are very very small and hard to read right now. Please also clarify, is the data in panel A the real data used to model the points in panels b-e? I would state this in the legend.

**References**

Tang et al., 2019
https://www.nature.com/articles/s41467-019-08640-0